# Identification of Hair Growth Promoting Components in the Kernels of *Prunus mira* Koehne and Their Mechanism of Action

**DOI:** 10.3390/molecules27165242

**Published:** 2022-08-17

**Authors:** You Zhou, Jingwen Zhang, Wanyue Chen, Xiaoli Li, Ke Fu, Weijun Sun, Yuan Liang, Min Xu, Jing Zhang, Gang Fan, Hongxiang Yin, Zhang Wang

**Affiliations:** 1College of Pharmacy, Chengdu University of Traditional Chinese Medicine, Chengdu 611137, China; 2College of Ethnomedicine, Chengdu University of Traditional Chinese Medicine, Chengdu 611137, China

**Keywords:** *Prunus mira* Koehne, alopecia, network pharmacology, vitamin E, β-sitosterol, linoleic acid

## Abstract

The application of the seed oil of *Prunus mira* Koehne (Tibetan name ཁམབུ།), a plant belonging to the *Rosaceae* family, for the treatment of alopecia has been recorded in *Jingzhu Materia Medica* (ཤེལ་གོང་ཤེལ་ཕྲེང་།) (the classic of Tibetan medicine) and *Dictionary of Chinese Ethnic Medicine*. This study aims to reveal the effective components and mechanism of hair growth promotion in the kernel of *Prunus mira* Koehne. Network pharmacology was used to predict the mechanism of action and effective components in the treatment of the kernel of *Prunus mira* Koehne. The contents of amygdalin in 12 batches of the kernel of *Prunus mira* Koehne were determined by HPLC. An animal model of the depilation of KM mice induced by sodium sulfide was created, and five effective components that promoted hair growth were initially screened. In the study of the effectiveness and mechanism of action, KM and C57BL/6 mice are selected as experimental objects, three screening tests for active components of the kernel of *P. mira* are performed, and three effective components are screened out from the eight components. HE staining was used to detect the number of hair follicles and the thickness of the dermis. RT-PCR and immunohistochemistry were used to evaluate the influence of the expression of indicators in the Wnt/β-catenin signaling pathway in skin, including β-catenin, GSK-3β, and mRNA and protein expression levels of Cyclin D 1 and LEF 1. The network pharmacology study showed 12 signaling pathways involving 25 targets in the treatment of alopecia by the kernel of *Prunus mira* Koehne. vitamin E (3.125 mg/cm^2^/d), β-sitosterol (0.061 mg/cm^2^/d), and linoleic acid (0.156 mg/cm^2^/d) in the kernel of *Prunus mira* Koehne can promote hair growth in mice, and the mechanism of action may be related to the Wnt/β-catenin pathway.

## 1. Introduction

*Prunus mira* Koehne, a plant belonging to the *Rosaceae* family, is named *P. mira* in Latin and ཁམབུ། in Tibetan (Kangbu, Holdkan) [1]. It is the direct origin of peach in Yunnan Province and the southwestern of Sichuan Province [2], and it is a rare “living Fossil Group” of peach genetic resources in the world. It has the characteristics of fast growth, long life, drought and cold tolerance. According to *Commonly Used Chinese Medicinal Materials Quality Sorting and Quality Research* (1997) [3], *P. mira* is distributed in the border areas of Sichuan Province, Yunnan Province and Tibet Autonomous Region, as shown in Figure 1. The main distinguishing points of *P. mira* and other species, such as *Amygdalus kansuensis* (Rehd.) Skeels and peach, are smooth surface of the pit, absence of holes, and has shallow and wide longitudinal grooves. Both *Jingzhu Materia Medica* (ཤེལ་གོང་ཤེལ་ཕྲེང་།) (the classic Tibetan medicine), compiled by the renowned Tibetan pharmacist Diamer Danzeng Pengcuo over roughly a period of 20 years, was finished in 1743 [4], and *Dictionary of Chinese Ethnic Medicine* record that the kernel oil of *P. mira* can darken and grow hair, it can treat alopecia. In addition, the *Dictionary of Chinese Ethnic Medicine* also records that the burnt ash of fruits of *P. mira* can be used to treat trauma and constipation, and grasserie [5]. According to *Sichuan Province Standard for Chinese Medicinal Materials* (2010) [6], its efficacy is consistent with that of *Amygdalus persica* L., which is used in traditional Chinese medicine. Nowadays, in addition to the kernel of *A. persica*. and *Amygdalus davidiana* (Carrière) de Vos ex Henry, some kernels of *P. mira* in the market are for sale [7].

Article 8 (j) of The Convention on Biological Diversity introduced the concept of “indigenous and local communities”. The Nagoya protocol on access and benefit sharing many of the core terms involves “indigenous and local communities” and requires users to obtain the “prior informed consent” of “indigenous and local communities” when accessing genetic resources and related traditional knowledge and share benefits equally with “indigenous and Local communities” [8]. Researchers should have ethnic and folk medicine knowledge from the people to serve the people’s idea [9]. Specimens of *P. mira* are collected from Sichuan and Yunnan provinces. The research group has good communication with local residents. The researchers have complied with the above conventions when conducting research of *P. mira*. The fast growth, strong adaptability, cold tolerance, longevity, and other features of *P. mira* have laid a good foundation for its development and utilization. It is a natural green organic food. At the same time, the fruit of *P. mira* has low sugar content and is rich in elements, such as Zn, Ca, and K, due to the special geography and climate of Tibetan areas. Compared with the fruits of *Amygdalus persica* L., the *P. mira* fruit has 2–3 times higher Zn content and more than 10 times vitamin C content [10,11]. It is a good original material for cultivating cold-resistant and long-lived peach varieties. The existence of plant cold-resistant genes can also provide huge genetic and hybrid value [12]. At present, GC-MS is used to identify the fat-soluble components of *P. mira*. There were 35 compounds, mainly oleic acid, β-sitosterol, *trans*-squalene, and vitamin E, with four unique compounds [13]. The contents of vitamin E, squalene, and β-sitosterol were determined by HPLC [14]. Oleic acid, linoleic acid, vitamin E, and other chemical components may have an effect on hair growth. However, the composition of the kernel of *Prunus mira* Koehne is complex, and the current control indicators are difficult to control its quality. Therefore, our research group determined the content of some of the chemical components related to hair growth.

Hair, an important accessory structure of the skin, protects the body. For example, hair can reduce the heat loss from the head [15]. The hair growth cycle can be divided into growth, regression, and resting periods. In the growth period, the cells at the bottom of hair follicles receive proliferation signals and extend and differentiate into the dermis to form hair follicles. During the growth period, the central column layer of the hair follicle first extends to the subcutaneous tissue and then grows back to form the hair shaft after reaching a certain depth. The growth period can be divided into six subperiods. The difference in the growth status of the hair follicles in the first five subphases is little, and the last subphase determines the length of the hair shaft. In some species, the hair follicles at the same location may have their own growth rhythms (such as humans and guinea pigs), and in some rodents (mice), the hairs at the same location are almost in the same hair cycle. At present, the factors affecting the hair follicle cycle include hair follicle damage, abnormal growth cycle, surrounding hair follicle growth cycle, and endocrine [16]. 

Alopecia is a skin disease characterized by hair loss [17] and is divided into seven levels in accordance with the area of hair loss. Studies found that its pathogenesis is related to heredity [18], endocrine [19], mental state [20]. It is pointed out that about 1/6 people in China have hair loss, among which 64% are men, and hair loss tends to be younger. Alopecia can lead to a decline in the quality of life [21], it has a greater impact on people’s appearance [22] and easily causes psychological burden [23,24]. Thus, the treatment of alopecia has become the focus of attention in the field of dermatology and the public. Nowadays, the drugs commonly used to treat hair loss are minoxidil and finasteride. Minoxidil can make hair grow and thicken and requires long-term medication. However, it is easy to relapse after stopping the drug [25]. Finasteride can inhibit the conversion of testosterone to dihydrotestosterone, thereby improving hair loss caused by strong androgen secretion. However, finasteride can easily cause sexual function problems [26]. The commonly used drugs for hair loss treatment in traditional Chinese medicine include *Fallopia multiflora* (Thunb.) Harald., *Ligustrum lucidum* Ait., *Platycladus orientalis* (L.) Franco., and *Polygonum multiflorum* Thunb. The commonly used traditional Chinese medicine compounds include traditional Chinese medicine hair growth liquid, Sangbai hair growth recipe, hair growth mixture of *Ginkgo biloba* L., and hair growth tincture [27]. In addition to drug treatment, plum blossom needle tapping, silver needle acupuncture, point injection, moxibustion, and other methods are commonly used to treat alopecia. The literature reported that the combination of vitamin E and plum blossom needle tapping is effective in treating alopecia areata [28]. Although the drugs and methods used in the clinical treatment of hair loss in modern medicine include minoxidil, finasteride, dutasteride, ketoconazole, prostaglandin drugs, laser therapy, hair transplantation [29,30], it is easy to relapse.

Network pharmacology is a technology that combines multiple disciplines such as systems biology, pharmacology, and computer science to mine and analyze drug targets [31]. Traditional Chinese medicine monomer and compound are the main application methods of modern Chinese medicine in the prevention and treatment of diseases. Therefore, the application of network pharmacology methods to study the relationship between the components of traditional Chinese medicine and the complex system of biological organisms is conducive to the discovery of the main compounds in traditional Chinese medicine that exert efficacy, determination of the molecular mechanism of drug prevention and treatment, and explanation of the mechanism of action of traditional Chinese medicine from a molecular perspective [32]. Network pharmacology is currently used to explain mainly the active components and mechanism of action of preparations or medicinal materials [33]. The traditional method of studying active components in Chinese pharmacy has laid a good foundation for the development of network pharmacology but has a large workload and low efficiency and is time-consuming. Existing research results showed that network pharmacology can reduce workload and save manpower and resources by simulating the interaction between drugs and the body [34]. Wu guosong [35] applied network pharmacology and found that compounds, such as geranidin, dihydrochelerythrine, and sesamin, contained in *Zanthoxylum nitidum* (Roxb.) DC. show potential anti-inflammatory activities due to the anti-inflammatory effects of COX-2 and MAPK14. The TNF signaling pathway is a potential signaling pathway for *Zanthoxylum nitidum* (Roxb.) DC. to achieve anti-inflammatory effects. Tao et al. [36] used network pharmacology to study the relationship between the prescriptions of *Curcuma aromatica* Salisb. recipe and found that the *C. aromatica* recipe may improve nutritional and metabolic diseases. The databases commonly used in network pharmacology can be divided into chemical and biological information-related databases. Commonly used chemical information-related databases are Traditional Chinese Medicine System Pharmacology Database and Analysis Ping (TCMSP), Taiwan Traditional Chinese Medicine Database (TCMD@Taiwan), and Traditional Chinese Medicine Comprehensive Database [37]. Commonly used biological information-related databases can be roughly divided into three categories. First, databases, such as Uniprot, Parmmaper, and Drug Bank, are used to find information about biological targets related to drugs or chemical components. Second, databases, such as OMIM and Genecard, are used to find disease-related targets. Third, databases, such as Sring, Mint, IntAct, are used to establish protein interaction relationships [38]. 

Previous studies found that the fat-soluble components of the kernel oil of *P. mira* within the range of 15.06–60.26 mg/cm^2^/d affect hair growth, but the effective components remain unclear [39]. The acute toxicity of the kernel oil of *P. mira* to mice, rats, and rabbits is studied, and results showed that the maximum doses of the kernel oil of *P. mira* administered orally to rats and mice are 144.612 and 289.224 g/kg/d, respectively. The safe maximum dose of the kernel oil of *P. mira* administered to rabbits through the skin is 482.28 mg/cm^2^/d [40]. The kernel of *P. mira* is safe for clinical use, and its fat-soluble components have no evident long-term toxicity on rats and irritation on the skin. However, the application of fat-soluble components on the scalp of the patient causes discomfort and cannot achieve the effect of clinical application. By looking for the main components or parts of the kernel of *P. mira* that exerts the medicinal effect, we can further clarify the mechanism of the kernel of *P. mira* promoting hair growth and improve the drug form for clinical use. This article first studied the effective components of the kernel oil of *P. mira* to promote hair growth. In accordance with previous research results and network pharmacology technology [39], eight kinds of components, including β-sitosterol and linoleic acid, are selected to evaluate the pharmacodynamics of promoting hair growth in KM and C57BL/6 mice and passed the pharmacodynamic evaluation screening. The components that have evident effects on promoting hair growth, namely, β-sitosterol, linoleic acid, and vitamin E, have also been initially explored. These three components in the kernel of *P. mira* are determined to improve quality and reveal the effective components of the kernel of *P. mira* promoting hair growth and its mechanism.

## 2. Material and Methods

### 2.1. Network Pharmacology Predicts the Mechanism and Active Components of the Kernel of P. mira in the Treatment of Hair Loss

Network pharmacology data collection first collected and sorted the chemical components of the kernel of *P. mira* through literature search, collected the components and the corresponding targets of the disease with the help of the database, and then matched the two targets to predict the mechanism of the kernel of *P. mira* treatment for alopecia for subsequent research and treatment. The mechanism of alopecia provides a reference. The database and analysis software used in network pharmacology are shown in Table 1. The chemical composition of the kernel of *P. mira* was comprehensively collected through modern biomedical literature databases, such as CNKI, VIP database, Pubmed, and Wanfang database, due to non-availability of data in TCMSP, TCMD@Taiwan, PubChem, and other databases. Drugs and disease-related targets were analyzed and screened. The chemical structure of its chemical components was searched in Pubchem database, 2D and 3D SDF files were downloaded, and then the Palmmapper database was used to collect target proteins of the components. In accordance with the synthesis of the Palmmapper database Score, genes with score >0.6 were selected. GeneCards was used to search the keywords “hair loss” and “alopecia” to search disease targets. The component-related targets of the kernel of *P. mira* and the alopecia disease-related targets were taken to intersect to obtain a common potential target and draw a Venn diagram. The relationship files among drugs, components, targets, and diseases were sorted out, and the Cytoscape software was used to draw a network diagram of drugs-components-targets-disease. Potential targets were imported into String, and score > 0.4 was selected to predict the interaction relationship between protein and protein. The degree of topological parameter was used to evaluate the importance of the target, and the top 30 Hub genes and PPI files were obtained. The PPI relationship files obtained by String were imported into the Cytoscape software to draw PPI diagrams. The Metascape database was used for Gene oncology (GO) enrichment analysis to find the biological processes, molecular functions, and cell composition of potential targets involved in the body. The Metascape database was used for KEGG enrichment analysis. 

### 2.2. The Contents of Amygdalin in the Kernel of P. mira Were Determined by HPLC

#### 2.2.1. Materials

Twelve batches of the kernel of *P. mira* were collected from Sichuan Province, Yunnan Province and other places in China. The collected specimens were identified as *P. mira* by Minru Jia, who was the professor of Chinese medicine identification at Chengdu University of Traditional Chinese Medicine. The voucher specimen was preserved in the college of ethnomedicine, Chengdu University of Traditional Chinese Medicine. HPLC analyses were carried out using Shimadzu LC-2030 high performance liquid chromatograph (Shimadzu, Kyoto, Japan), and the purity by HPLC detection was greater than 97%. The CPA2250 electronic balance was purchased from Sartorius, Goettingen, Germany. The reference substances of amygdalin (batch number PS20190922) were purchased from Chengdu Pusi Biotechnology Co., Ltd., (Chengdu, China). The purity by HPLC detection was greater than 97%. Acetonitrile (chromatographically pure) was purchased from Anhui Tiandi High Purity Solvent Co., Ltd., (Anqing, China), and methanol (analytical pure) was purchased from Sigma-Aldrich Trading Co., Ltd., (Saint Louis, MO, USA), and water was ultra-pure water, and other reagents were analytically pure.

#### 2.2.2. Sample Preparation

Desheng oil press was used to extract seeds at 40 °C. The optimum extraction conditions were obtained by experiments of different temperatures, different solvents, different extraction times, different extraction volumes, and different extraction methods. The preparation method of the test sample of amygdalin was to accurately weigh 0.5 g of the kernel of *P. mira*, add 25 mL of petroleum ether, ultrasonic (200 W, 40 KHz) for 30 min, and filter out the petroleum ether, add 25 mL of methanol solution, ultrasonically extract for 30 min, filter, transfer the filtrate to a 25 mL volumetric flask, make constant volume of methanol, and pass through a 0.45 μm microporous membrane [41].

#### 2.2.3. HPLC Conditions

Shimadzu LC-2030 HPLC (Shimadzu, Kyoto, Japan) was used to determine the content of amygdalin, and the processing of chromatogram and the calculation of peak area was automatically integrated by the computer using LabSolution software (VersionDB, Shimadzu, Kyoto, Japan). The content of amygdalin was determined by HPLC, and different detection wavelengths (203, 210, 220, 230, 242 nm), mobile phase (acetonitrile: water = 30:70, 20:80, 10:90), column temperature (25 °C, 30 °C, and 35 °C), and flow rate (0.8, 1.0, 1.2 mL/min), according to the peak resolution, peak shape, peak area, etc., the best chromatographic conditions were determined. In the preparation of the test product, the following conditions were considered to ensure the best test conditions, such as the extraction method (cold soaking for 12 h, heating and refluxing for 30 min, and ultrasonic for 30 min), extraction solvent (50% methanol, 70% methanol, methanol solution), extraction volume (5, 15, 25 mL), extraction time (15, 30, 60 min). 

#### 2.2.4. Preparation and Calibration of Standard Solutions

Amygdalin was accurately weighed 6.4001 mg, and the volume was fixed into 5 mL volumetric flasks. The concentrations of the reference substance stock solution were 1280.00 µg/mL. The stock solutions of the reference substance of amygdalin were diluted two times in sequence, the concentration of amygdalin was 1280.00, 640.00, 320.00, 160.00, 80.00 µg/mL. A total of 10 μL of sample was taken to determine the peak area, and draw the standard curve equation of amygdalin.

### 2.3. Screening of the Effective Components in the Treatment of Two Depilatory Model Animals with Chemical Components in P. mira

#### 2.3.1. Animals and Test Sites

430 KM mice (SPF grade, 20 ± 2 g, male) and 170 C57BL/6 mice (SPF grade, 6 weeks old, 20 ± 2 g, male and female) were provided by Chengdu Dashuo Experimental Animal Co., Ltd (Chengdu, China). The experimental animal production license number were SCXK (Sichuan Province) 2015-030 and SCXK (Sichuan Province) 2015-030, respectively, and the numbers of experimental animal quality certificates were 51203500006767 and 51203500009764, respectively. All animals were raised in the Science and Technology Building of Chengdu University of Traditional Chinese Medicine. The license number of animal use was SYXK (Sichuan Province) 2020-124. Experiments were carried out in the Ethnomedicine Resource Evaluation Laboratory of Chengdu University of Traditional Chinese Medicine (third-level scientific research laboratory of the State Administration of Traditional Chinese Medicine, No. TCM-2009-320). The abovementioned experimental animals had been reviewed and approved by the Experimental Animal Ethics Committee of Chengdu University of Traditional Chinese Medicine. Its approval number was 2018-23.

#### 2.3.2. Preparation of Experimental Samples

Vitamin E soft capsules (batch number 10891003) were purchased from Xingsha Pharmaceutical (Xiamen) Co., Ltd., (Xiamen, China), with National Medicine Standard H35020242. β-sitosterol (batch number AF8102901), *trans*-squalene (batch number AF8091427), oleic acid (batch number AF8062708), campesterol (batch number AF8112891), fucosterol (batch number AF8112891), and amygdalin (batch number AF8051847) were purchased from Chengdu Alpha Biotechnology Co., Ltd., (Chengdu, China). Linoleic acid (batch number Y-096-170426) was purchased from Chengdu Pusi Biotechnology Co., Ltd., (Chengdu, China), Minoxidil liniment (batch number 1805-932) was purchased from Sichuan Meidakang Huakang Pharmaceutical Co., Ltd., (Deyang, China). with National Medicine Standard H20052642 and used as a positive control drug [42].

Clinically, the concentration of minoxidil liniment was 20 mg/mL at 0.1 mL each time once a day. The area of hair removal was 4 cm^2^, and the dosage was 0.5 mg/cm^2^/d. The dose setting method of the test substance was the same as above. The method was performed in accordance with previous studies [39]. In accordance with previous studies [43], peanut oil was used as a solvent for drugs with low polarity, and physiological saline was used for drugs with high polarity. Pre-experiments were conducted on peanut oil and physiological saline. Peanut oil and physiological saline did not affect the efficacy. Five doses were set for each tested drug (Table 2). The corresponding medicine was accurately weighed, placed in a bottle, and added with solvent. The mixture was shaken with a vortex instrument until complete dissolution or even dispersion was achieved. The prepared medicine was stored in a refrigerator maintained at 4 °C.

#### 2.3.3. Preliminary Screening of Eight Components in the Kernel of *P. mira* in the Treatment of KM Mice after Depilation Induced by Sodium

430 KM mice were randomly divided into 43 groups, including blank control group, model control group, minoxidil control group, and eight different test drug groups (each test drug had five doses), after stratification by body weight. Ten animals were placed in each group and reared in two cages. Considering that 430 KM mice were needed for eight components and the number of animals was large, the experiment was carried out in batches. Each batch of experiments involved each drug and subgroup of animals. The mice in the blank control group only shaved their back hair by about 2 × 2 cm, and mice in other groups first shaved the back hair of the same area as the mice in the blank control group. Na_2_S (6%) was evenly applied on the shaved area and washed off after about 2 min. Mice with smooth and undamaged skin was selected for follow-up experiments [44]. Exactly 24 h after making the model, the drug solution was evenly applied to the depilatory area and was allowed to absorb. Mice were observed to prevent them from licking their backs. After the drug solution dried, mice were returned to the cage. Once a day for 7 consecutive days, before each administration, the back area was washed with physiological saline (0.1 mL each time) and no drug was applied before drying [45]. On the second, fourth, and sixth days, the depilated parts of the mice were graded according to the standard [46] (Table 3). After 24 h of the last administration, 5 hair were plucked to measure the hair length of the mice in the middle of the administration area, and the average hair length was measured (cm) [47]. After the animal was sacrificed, the same position was used by a 7 mm punch to take a piece of skin, and all the hair on the skin were scraped off and weighed [48].

#### 2.3.4. To Verify the Effect of Five Active Components in the Kernel of *P. mira* on C57BL/6 Mice after Depilation Induced by Sodium

170 C57BL/6 mice were randomly divided into 17 groups in accordance with body weight. Ten animals were included in each group and reared in two cages. According to the literature [49] and preliminary experiments, C57BL/6 mice needed external stimulation, such as plucking and sodium sulfide, to enter the growth phase from the resting phase to the hair follicles. Therefore, all experimental mice should be treated with Na_2_S. There was no blank control group in this experiment. Given that not all 170 mice were in the resting phase (skin was pink), only mice in the resting phase could be modeled and administered. Considering the operability of the experiment, mice were tested in batches. Each batch involved each drug and subgroup of animals. The time when the skin color changed from pink to black was recorded, and the result was rated and photographed on days 7, 14, and 21. Grading standards are shown in Table 2. Considering that the C57BL/6 mice gradually developed hair growth after seven days of administration, hair length was measured on days 14 and 21. The method was the same as the above, and the new hair weight was measured as above. About 0.5 cm^2^ C57BL/6 mouse back skin tissues of each group were taken for subsequent HE, immunohistochemistry, and RT-PCR detection.

### 2.4. Mechanism of the 3 Active Components in the Kernel of P. mira in the Treatment of C57BL/6 Mice Depilation Model Induced by Sodium Sulfide

#### 2.4.1. Grouping and General Indicators

Through comprehensive analysis of the results of skin color change time, hair growth status, hair length, and weight of C57BL/6 mice, β-sitosterol group 2 (0.061 mg/cm^2^/d) and the second group of linoleic acid group 2 (0.156 mg/cm^2^/d), vitamin E group 2 (3.125 mg/cm^2^/d) had a better effect on promoting hair growth compared with the others. Thus, one dose group for each of the three components was selected for the mechanism of action.

#### 2.4.2. Skin Histological Observation

On the 22nd day, the skin of the administration area of C57BL/6 mice was fixed, and the inspection and image collection were performed according to the pathological examination SOP procedure. The number of primary hair follicles, the number of secondary hair follicles, the total number of hair follicles and the thickness of the dermis were observed and counted, and the average value was calculated [50].

#### 2.4.3. RT-PCR Detection of mRNA Expression of Four Targets in the Skin Wnt/β-catenin Pathway

On the 22nd day, the skin of the depilated area on the back of the C57BL/6 mice was collected, and total RNA was extracted from the dorsal skin tissue by using the RNA extraction kit (RE-03014, Forgene Biotechnology Co., Ltd., Wuhan, China). Total RNA was reverse-transcribed using the PrimeScript RT reagent kit (RR047A; Bio-Biology, Dalian, China) in accordance with the manufacturer’s recommendations. Reagents were added to the kit in sequence at 42 °C for 2 min. Genomic DNA removal reaction and reverse transcription reaction on the PCR machine are shown in Table 4 and Table 5, respectively. The full sequence of the gene was searched in the NCBI database. The Primer Premier primer design software was designed to screen specific primers for each gene and purified using ULTRAPAGE (Table 6). The real-time fluorescence quantitative PCR reaction is shown in Table 7 and Table 8. The β-actin of the sample skin tissue and the dissolution curves of the four indicators were all single peaks, which indicated that the amplified product was the target product of the corresponding gene and that primer dimers and nonspecific products were not observed (Figure 2a). The amplification curves of the tested genes were smooth and could enter the plateau phase, which indicated that the PCR reaction system and reaction program were set reasonably (Figure 2b). The CT value of each test sample in the PCR process was analyzed, and the relative mRNA expression of X was calculated by 2^−ΔΔCT^.

#### 2.4.4. Immunohistochemical Method to Detect the Protein Expression of Four Targets in the Skin Wnt/β-catenin Pathway

On the 22nd day, the skin of the depilated area on the back of the mice was fixed and added dropwise with blocking solution at room temperature for 20 min. The pretreated skin was added with GSK-3β antibody (1:200), rabbit β-catenin antibody (1:300), rabbit Cyclin D 1 antibody (1:50), and rabbit LEF 1 antibody (1:100), rinsed, and incubated overnight at 4 °C. The treated skin was then dripped with biotinylated goat anti-rabbit/mouse antibody, incubated at 37 °C for 30 min, washed three times with PBS, developed at room temperature, counterstained, mounted, and finally checked and calculated the optical density. 

### 2.5. Statistical Methods

The measurement data were compared between groups by “independent sample *t*-test” in SPSS 17.0 software(International Business Machines Corporation, New York, NY, USA), which was expressed by “mean ± standard deviation ($\bar{x}$ ± s)”. Data were analyzed by the Mann–Whitney test in the SPSS 17.0 software for comparison between groups. A single asterisk (*, *p* ≤ 0.05) indicated a statistical difference between the averages, and a double asterisk (**, *p* ≤ 0.01) indicated a high significant difference between the averages.

### 2.6. Principal Component Analysis Method

The SIMCA-P+ software (Version 14.1, MKS Data Analytics Solutions, Umea, Sweden) was used to perform principal component analysis. A total of 16 measurement indicators of the hair loss model (total number of hair follicles; number of primary hair follicles; number of secondary hair follicles; dermal thickness; and Cyclin D 1, β-catenin, GSK-3β, and LEF 1 mRNA and protein expression levels) were included. Five groups, i.e., model control group, minoxidil control group, β-sitosterol group 2, vitamin E group 2, and linoleic acid group 2, were used in this experiment.

## 3. Results and Discussion

### 3.1. Network Pharmacology

In accordance with the literature [13], a total of 17 chemical constituents of the kernel of *P. mira* were screened (Table 9). The SDF files of chemical components sought in the PubChem database were imported one by one into the Pharmmapper database for scoring. After the duplicate was deleted, the final corresponding targets were 5085, and targets with score higher than 0.6 were selected. After the duplicate was deleted, 216 targets were obtained. The GeneCards library was searched to obtain 9640 disease targets. After matching the two related targets, 149 potential targets for the treatment of alopecia diseases were obtained, and the Venn diagram was drawn (Figure 3 and Table 10). The network diagram of “the kernel of *P. mira*-components-Target-Disease” was established using the Cytoscape software (Version 3.7.1, Institute for Systems Biology, Seattle, WA, USA) (Figure 4).

The network included 168 nodes and 842 edges. The relationship between the kernel of *P. mira*, chemical composition, potential target, and hair loss is demonstrated by different colors and shapes. Then, the “protein–protein” network topology was analyzed, and the PPI network was constructed using the String website (https://String-db.org; accessed on 23 May 2020) and Cytoscape, as shown in Figure 5. A large circle in the figure indicated an important gene. The degree of freedom was used as the selection condition for the hub gene. A high degree of freedom indicated high possibility that the kernel of *P. mira* will function through the target gene. The top 30 degrees of freedom were regarded as the hub gene, as shown in Figure 6. 

The biological process and molecular function analyses of targets of the kernel of *P. mira* were carried out, and the first 20 processes with the least significance were selected (Figure 7a). GO analysis showed that heme binding, tetrapyrrole binding, nuclear receptor activity, transcription factor activity, steroid hormone receptor, S100 protein binding, damaged DNA binding, hydrolase activity, vitamin binding, fatty acid derivative binding, and antioxidant activity, might be related to alopecia. KEGG analysis was performed on 149 potential targets, and a total of 89 signal pathways were obtained, of which 12 pathways were related to alopecia (Figure 7b). The thyroid hormone (hsa04919) and HIF-1 (hsa04066) pathways were related to angiogenesis. The Rap1 (hsa04015) and AGE-RAGE (hsa04933) pathways were related to inflammation. The p53 (hsa04115) and FoxO (hsa04068) pathways were related to death. The PI3K/Akt (hsa04151), Wnt (hsa04310), and TGF-beta (hsa04350) pathways were related to cell proliferation and differentiation. The estrogen (hsa04915), neurotrophin (hsa04772), and cAMP (hsa04024) pathways were related to lipid metabolism. Figure 8 shows that 25 of the 149 potential targets were related to these signal pathways. Among these targets, *CREBBP*, *EIF4E*, *SOD2*, and *CYCS* had high degrees of freedom, and the 25 targets were predominantly derived from 13 components. According to the results of previous research and literature review, stimulation of Wnt signaling pathway can regulate the growth of hair follicles [51]. In this paper, the Wnt signaling pathway was selected as the main research object, and five related targets, namely, *CREBBP 1, CCND1, RAC1, CUL1,* and *EP300* were found. A total of 13 chemical components, such as *trans*-squalene, vitamin E, fucosterol, linoleic acid, oleic acid, and campesterol, were related to these five targets. Thus, it may be a potential effective ingredient of *P. mira* to promote hair growth, but the results of network pharmacology still need further experimental verification.

The pathways predicted by network pharmacology are related to angiogenesis, inflammation, cell proliferation, and apoptosis, which are consistent with the factors currently considered to affect hair loss, such as inflammation, follicular cell cycle, and follicular blood circulation. The literature review showed that the mechanism of action for the treatment of alopecia is mostly concentrated in the Wnt and EGFR signaling pathways, and this finding is similar to the predicted results of network pharmacology. Current research on the Wnt pathway predominantly involves regulating the proliferation of breast cancer, colon cancer, and other cancer cells [52], regulating osteoarthritis [53] and osteoblast differentiation [54], participating in metabolism [55], and regulating hair follicles, and hair regeneration [56]. The PI3K/Akt pathway is also one of the signaling pathways predicted by network pharmacology. It is one of the classic pathways of the EGFR signaling pathway [57]. It can induce or inhibit cell apoptosis [58]. The current research focuses on tumors [59], anti-inflammatory [60], osteoblast proliferation and differentiation [61], and growth factor-mediated neuronal activity [62], but there are also reports that the PI3K/Akt pathway is related to hair growth [63]. A crosslink between the Wnt and EGFR signaling pathways may be present [64]. In the breast epithelial cell line HC11, Wnt can affect the EGFR signaling pathway through Wnt5a, Wnt1, and other cell membrane ligands [65]. Both Wnt and EGFR pathways are involved in Drosophila. Polarity of eye cells is formed [66]. From the perspective of research direction and function, these pathways are also similar. Crosslinking between Wnt and EGFR signaling pathways may be present in hair follicle regulation. Teng et al. found that the EGFR and EGF signals in the hair follicles of mice with mutations in the key genes for hair loss are weakened, and the continued high expression of EGF in mice can keep the hair cycle in the growth phase [16]. Yi et al. found that β-catenin can regulate the proliferation of hair follicle stem cells by regulating the PI3K/Akt pathway [67], indicating that the PI3K/Akt signaling pathway is related to the Wnt signaling pathway. 

### 3.2. HPLC Analysis

#### 3.2.1. Optimization of Chromatographic Separation Conditions

Shimadzu LC-2030 HPLC (Shimadzu, Japan) was used to determine the content of amygdalin, and the processing of chromatogram and the calculation of peak area was automatically integrated by the computer using LabSolution software(VersionDB, Shimadzu, Kyoto, Japan). The optimum chromatographic conditions were obtained by investigating the chromatographic conditions of different wavelengths, different mobile phases, different column temperatures, and different flow rates. The chromatographic column is Hypersil ODS2 (250 × 4.6 mm, 5 μm) (Thermo Fisher Scientific, Waltham, MA, USA). When determining amygdalin, the column temperature was 30 °C, and the flow rate was 0.8 mL/min, and the injection volume was 10 µL, and the detection wavelength was 203 nm, and the mobile phase was acetonitrile-water (80:20). The best extraction method of amygdalin in sample solution was 0.5 mol/L potassium hydroxide methanol and the extraction volume is 25 mL, the extraction method was ultrasonic and the extraction time was 30 min.

#### 3.2.2. Quantitative Analysis of Amygdalin in the Kernel of *P. mira*

Amygdalin showed good linear relationships in the range of 80.00–1280.00 µg/mL. The methodological investigation RSD was less than 3%, it was indicated that the method is reliable. See Figure 9 and Table 11. In 12 batches, the content of amygdalin in Xinlong County, Jiulong County and Kangding City was significantly higher than that of other producing areas. There was no significant difference in amygdalin content in the kernel of *P. mira* from other producing areas. See Table 12 for details.

### 3.3. Dose-Effect Relationship of Eight Chemical Components in the Kernel of P. mira to Promote Hair Growth

KM mice with depilation induced by sodium sulfide was used to evaluate the efficacy of eight components (Figure 10) (i.e., β-sitosterol, linoleic acid, vitamin E, oleic acid, *trans*-squalene, campesterol, fucosterol, and amygdalin) of the kernel of *P. mira*. β-sitosterol groups 2, 3, 4, and 5; vitamin E groups 1 and 2; linoleic acid group 2; fucosterol group 3; and amygdalin groups 1 and 3 could increase the grade of newborn hair condition of mice on the fourth day. Amygdalin group 1 and fucosterol group 3 could increase the hair length of mice. Vitamin E groups 1, 2, and 3 and β-sitosterol group 2 could significantly increase the weight of new hair (Table 13 and Figure 11). After considering the above results comprehensively, vitamin E, β-sitosterol, linoleic acid, amygdalin, and fucosterol had evident promoting effects. Therefore, 3 doses of these 5 components, i.e., β-sitosterol groups 2, 3, and 4 (0.061, 0.031, and 0.016 mg/cm^2^/d); linoleic acid groups 1, 2, and 3 (0.313, 0.156, and 0.078 mg/cm^2^/d); vitamin E groups 1, 2, and 3 (6.125, 3.125, and 1.563 mg/cm^2^/d); amygdalin groups 2, 3, and 4 (0.061, 0.031, and 0.016 mg/cm^2^/d); and fucosterol groups 2, 3, and 4 (0.061, 0.031, and 0.016 mg/cm^2^/d), were selected for subsequent sodium sulfide-induced hair loss model in C57BL/6 mice for re-evaluation of the efficacy. 

### 3.4. Re-Evaluating of the Effect of Five Active Components in the Kernel of P. mira in Promoting Hair Growth

C57BL/6 mice were used to re-evaluate the effectiveness of the five active components in the kernels of *P. mira*. β-sitosterol groups 2 and 4, linoleic acid group 2, and vitamin E group 2 could remarkably shorten the time for the skin to darken and increase the status of new hair and hair length and weight. In addition, β-sitosterol group 3, linoleic acid group 1, vitamin E group 3, amygdalin group 2, and fucosterol group 2 could significantly increase the hair length and weight of mice on day 14 (Table 14 and Table 15). The use of heat maps could show the differences between the data intuitively. The 17 groups (i.e., model control group; positive control group; β-sitosterol groups 2, 3, and 4; linoleic acid groups 1, 2, and 3; vitamin E groups 1, 2, and 3; fucosterol groups 2, 3, and 4; and amygdalin groups 2, 3, and 4) of 7 efficacy index data (i.e., skin darkening time; 7-, 14-, and 21-day newborn hair growth rating; 14- and 21-day hair length; and hair weight) were normalized to draw a heat map. As shown in Figure 12, β-sitosterol, linoleic acid, and vitamin E had a significant promoting effect on mice hair loss induced by sodium sulfide. According to the statistical results of the number of days of skin darkening; the status, length, and weight of new hair; and results of heat map analysis, the three dose groups of β-sitosterol, vitamin E, and linoleic acid could make hair follicles grow in advance, indicating a certain promoting effect on hair growth. Therefore, β-sitosterol group 2 (0.061 mg/cm^2^/d), linoleic acid group 2 (0.156 mg/cm^2^/d), and vitamin E group 2 (3.125 mg/cm^2^/d) were selected for observing the effect of drugs on the number of hair follicles, dermal thickness, and Wnt/β-catenin pathway.

### 3.5. Mechanism of the Main Active Components in the Kernel of P. mira to Promote Hair Growth

Through the comprehensive analysis of the results of skin color change time, newborn hair growth rating, and newborn hair length and weight of C57BL/6 mice, the β-sitosterol group 2 (0.061 mg/cm^2^/d), linoleic acid group 2 (0.156 mg/cm^2^/d), and vitamin E group 2 (3.125 mg/cm^2^/d) had a good effect on promoting hair growth (See Figure 13). Thus, one dose group for each of the three components was selected for the mechanism of action. The number of hair follicles and skin dermal thickness of mice in the three groups were tested, and the effects of the three groups of doses on Wnt/β-catenin signaling pathway were studied (Table 16 and Figure 14). No significant effect was observed on the thickness of the dermis and the number of hair follicles, but the three groups could upregulate the mRNA expression of LEF 1 and GSK-3β in the Wnt/β-catenin signaling pathway and had a significant upregulation trend on the mRNA expression of β-catenin (Table 17). All three groups had no significant effect on the protein expression of the four indicators (Table 18 and Figure 15).

The SIMCA-P (Version 14.1) software was used to perform principal component analysis (PCA-X) on 16 indicators in five groups and analyze the control intensity of the mechanism of action, and five principal components were extracted with R2X = 80.5%, which indicated that the top five principal components could reflect 80.5% of the original data. The first two components were extracted to obtain Figure 16a. The model and the positive control groups were at both ends of the Y axis. The samples of the three administration groups were closer to those of the positive control group, indicating that the hair loss of mice in the administration group tended to improve. A trend of improvement was observed. In addition, the groups in the area of the circle followed the order: β-sitosterol group 4 > linoleic acid group 2 > vitamin E group 2. The strength of the hair growth-promoting effect was also inversely proportional to the area of the circle. PCA-X found that the three groups had a certain improvement effect on the alopecia of mice caused by sodium sulfide (Figure 16a,b).

Lipids can participate in the development and regulation of hair and skin [68]. Linoleic acid, β-sitosterol and vitamin E are all fat-soluble components. Truong et al. [69] found that oil of red ginseng (*Talinum paniculatum* (Jacq.) Gaertn.) and its main components, linoleic acid and β-sitosterol, can promote hair growth by regulating the Wnt/β-catenin signaling pathway. Upadhyay [70] found that β-sitosterol gel and β-sitosterol-phospholipid complex improved androgenic alopecia by inhibiting 5α-reductase. Linoleic acid derivatives can maintain the stability of hair follicles by maintaining the lipid metabolism within the hair and hair follicles [71] to keep the hair in a normal growth state [72]. The subcutaneous injection of vitamin E in patients with alopecia areata or plum blossom needle tapping combined with vitamin E rubbing can significantly improve the hair growth of patients with alopecia areata [73].

The Wnt/β-catenin signaling pathway is one of the important pathways related to hair loss. This pathway is a multichannel signal transduction pathway and participates in the process of embryonic cell development, division and differentiation, and tissue regeneration. This pathway is highly conserved in genetics, and the Wnt pathway among different species is similar. The β-catenin protein is a key factor in activating the pathway [74] and can be phosphorylated by GSK3β and then degraded [75]. However, when it accumulates to a certain amount, it is transferred from the cytoplasm to the nucleus and interacts with TCF/LEF family transcription factors (LEF 1) and binds to initiate transcription and translation of target genes, including C-myc, Cyclin D 1, etc., [76] (Figure 17). β-Catenin, GSK3β, LEF 1, Cyclin in the downstream of the Wnt/β-catenin pathway D 1 are the four important factors to be studied.

Studies found that β-sitosterol, linoleic acid, and vitamin E can upregulate the LEF 1 and the mRNA expression levels of GSK-3β, β-catenin, and Cyclin D 1 in the Wnt/β-catenin signaling pathway. The expression levels of mRNA and protein of Cyclin D 1 have no evident influence. The increase of LEF 1 mRNA expression and the rising trend of β-catenin mRNA expression indicate that the Wnt/β-catenin pathway is in an activated state. However, the mRNA expression of GSK-3β increases. This result is consistent with a previous study on the effect of light walnut oil on GSK. The effect of GSK-3β is consistent. The expression of GSK-3β may be regulated by other signal pathways, such as GSK-3β-MCL-1 [51], but further experimental verification is needed. Immunohistochemistry results showed that vitamin E, β-sitosterol, and linoleic acid have no evident effect on the protein expression of GSK-3β, β-catenin, Cyclin D 1, and LEF 1, which may be because the skin of mice in the administration group has changed from black to gray and pink to enter the resting phase after the third week. Although Wnt/β-catenin pathway is activated, its activity is reduced and hair growth is not observed in some mice [77]. Thus, no statistical difference in overall protein expression is observed. Mouse skins at different administration periods should be selected for testing in subsequent experiments. In this article, the expression levels of Cyclin D1 mRNA and protein do not change significantly. According to literature reports [78], it was found that the hair of mice knocked out of Cyclin D 1 gene grew normally, and that Cyclin D 1 gene had no obvious effect on the proliferation of hair follicle cells. However, other literatures indicate that Cyclin D 1 is related to hair follicle stem cells [79] and that the relationship between Cyclin D 1 gene and related signal pathways and hair follicles should be studied deeply. In addition, interactions between Wnt/β-catenin pathway and EGFR signaling pathway in the regulation of hair growth may be present. Signal pathways do not exist independently but are connected to each other to form a complex network. However, research on the relationship between the signal pathways remains lacking, suggesting that attention should be paid to the study of the Wnt/β-catenin pathway and the EGFR signal pathway in subsequent experiments. The connection and interaction of the hair loss treatment, the mechanism network of the treatment of hair loss, and the mechanism of its function are clearly revealed.

The commonly used hair loss agents in the current literature include hair removal cream, cyclophosphamide, rosin and paraffin, testosterone propionate, and sodium sulfide. The first three reagents are not targeted at specific types of hair loss. Cyclophosphamide is predominantly used for patients with hair loss caused by chemotherapy, and testosterone propionate is used for androgenetic alopecia. However, these hair loss models have causes that cannot fully simulate human hair loss [80]. Because the literature found that hair removal animal models are commonly used in C57BL6 mice, KM mice, and SD rats, and mostly female, so two different mice were selected in the experiment. Animal experiments of a variety of different species are conducive to evaluating the effects of drugs and avoiding false positive results. KM mice have a short hair growth cycle, which is suitable for short-term and rapid screening. C57BL/6 mouse skin has evident periodicity, and long hair growth cycle is suitable for long-term efficacy evaluation. C57BL/6 mice are in the growth phase when the skin is black. In the growth phase, secondary hair embryos are activated to differentiate into hair follicles. The cells of hair follicles are highly active and differentiate to form pigmented hair shafts. Hair follicle metabolism decreases during the anagen phase, and cells gradually undergo apoptosis. The skin color is gray at this time. At the end of the anagen phase, the cells of the hair follicle stop moving, and only a layer of differentiated cells remains on the surface of the hair papilla. The hair follicle forms a secondary hair embryo and enters the resting stage, and the skin of C57BL/6 mice turns pink [81]. Given this feature, C57BL/6 mice are widely used in hair growth animal experiments and skin disease-related fields [82]. In addition, when using C57BL/6 mice for experiments, all mice need to use sodium sulfide for hair removal. This is because the hair follicles of C57BL/6 mice are in the resting phase under normal physiological conditions and must rely on huge external stimuli (sulfide sodium treatment and plucking) can make the hair follicles of C57BL/6 mice enter the growth cycle [83]. Therefore, C57BL / 6 mice were used in this study, and no blank control group was established. The preliminary evaluation of efficacy using KM mice and re-evaluation of efficacy in C57BL/6 mice found that β-sitosterol, linoleic acid, and vitamin E allowed mice to enter the hair growth phase early, thus increasing the length and weight of new hair. The effect of these components in promoting hair growth is more evident than those of others. Although KM and C57BL/6 mice are used in this article, they still have limitations. The animal hair follicle cycle is different from the human hair growth cycle, and hair follicles are collected after the experiment is completed. Changes in some related proteins during the growth cycle have not been discovered. Thus, experimental models that are in line with the causes of human hair loss should be explored, and animals that are close to the human hair growth cycle for experiments should be determined. Through clinically consistent models, various animal experiments, multiple-time point sampling analysis, and reasonable index evaluation methods reflect the effect of drugs on hair loss.

At present, the fact that the kernels of *P. mira* are not included in the *Pharmacopoeia of the People’s Republic of China* (2020) has become a bottleneck restricting the cultivation and use of the kernels of *P. mira*, and the non-receipt by pharmaceutical factories has led to a sharp decline in usage. However, in the eyes of sellers in the medicinal material market, its appearance quality seems to be no weaker than that of peach kernels (the kernels of *Prunus persica* (L.) Batsch and *Prunus davidiana* (Carr.) Franch.). The author visited the medicinal material market in Chengdu’s International Trade City and found that the drug sellers were willing to buy the kernel of *P. mira* at a price of about 40 Yuan/kg. The peach kernels in the *Pharmacopoeia of the People’s Republic of China* (2020) are derived from *Prunus persica* (L.) Batsch and *Prunus davidiana* (Carr.) Franch. [84] and has the functions of activating blood circulation, removing blood stasis, and moistening intestines and defecating. Peach kernels are distributed in Hebei, Shaanxi, Yunnan, Gansu, and Sichuan Provinces [85]. At present, HPLC is used to study the fingerprints of the kernel of *P. mira* and peach kernels from many producing areas, and HPLC results of peach kernels show 12 common peaks and 3 different peaks, indicating that the chemical components of the two are roughly similar [86]. In the future, various experimental animals, such as mice, rats, and rabbits, can be used to create models of coagulation and bleeding, constipation, and gastrointestinal tract movement, to evaluate the effectiveness of the two systems and combine quality control indicators and pharmacodynamic indicators. The logistic algorithm evaluates and grades the quality of both [87]. In addition, the safety of acute toxicity in rats and mice and long-term toxicity in rats should be evaluated. The similarities and differences of quality control indicators, safety evaluation and system effectiveness evaluation were comprehensively evaluated, which laid the foundation for its entry into the *Pharmacopoeia of the People’s Republic of China* (2020). If the kernel of *P. mira* can be included in the *Pharmacopoeia of the People’s Republic of China* (2020), the acquisition, planting, and processing of the kernel of *P. mira* can be promoted in Tibetan areas, and the local medicinal resources especially for the poverty alleviation in the deeply impoverished areas of the Tibetan area can be fully used.

## 4. Conclusions

Network pharmacology showed 149 targets for the chemical composition and alopecia in the kernel of *P. mira* and constructed the “protein–protein network interaction diagram” and the network diagrams of “components in the kernel of *P. mira*-chemical-targets-hair loss”. KEGG enrichment results showed 12 alopecia-related pathways, involving 25 targets and 13 components. The contents of amygdalin in 12 batches of the kernel of *Prunus mira* Koehne were determined by HPLC, which improved the quality standard. Furthermore, vitamin E (3.125 mg/cm^2^/d), β-sitosterol (0.061 mg/cm^2^/d), and linoleic acid (0.156 mg/cm^2^/d) in the kernel of *P. mira* can promote hair growth in mice, and its mechanism of action may be related to the Wnt/β-catenin pathway. Research results provided a basis for revealing the pharmacodynamic material basis and mechanism of the kernel of *P. mira* for the treatment of alopecia and clues for the development of new drugs for the treatment of alopecia.

## Figures and Tables

**Figure 1 molecules-27-05242-f001:**
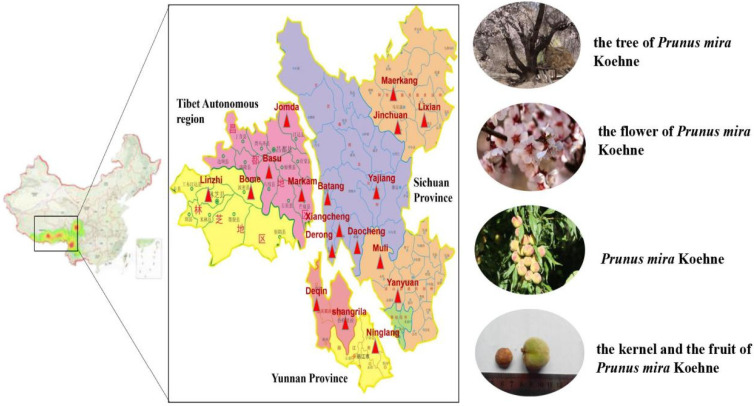
The distribution of *P. mira* in China.

**Figure 2 molecules-27-05242-f002:**
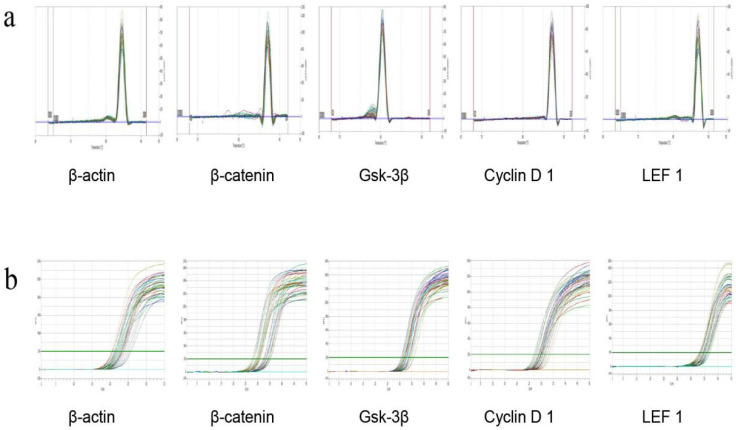
(**a**): The dissolution curve of mRNA RT PCR products of four indicators in mice skin tissue. (**b**) RT-PCR product amplification curves of four indicators in mice skin tissue.

**Figure 3 molecules-27-05242-f003:**
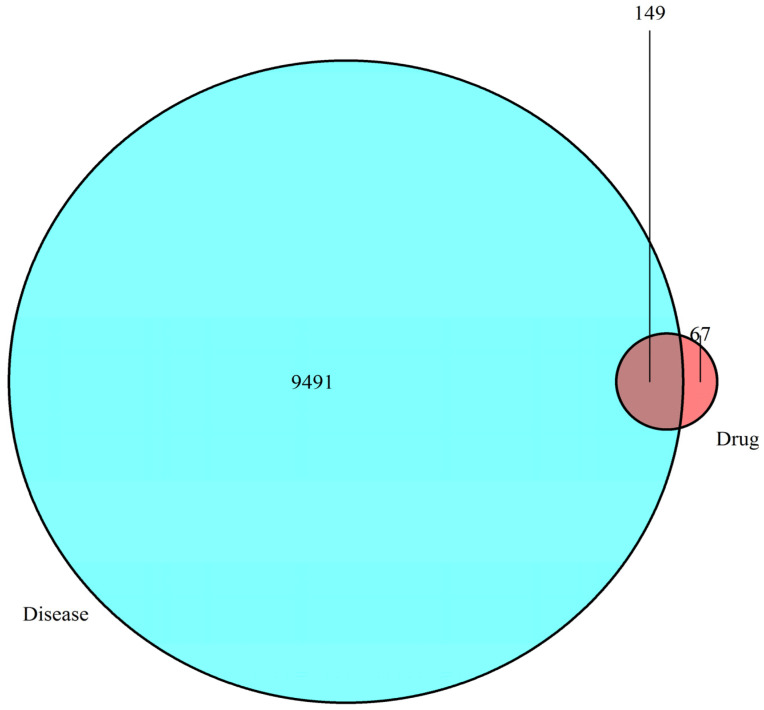
Venn diagram.

**Figure 4 molecules-27-05242-f004:**
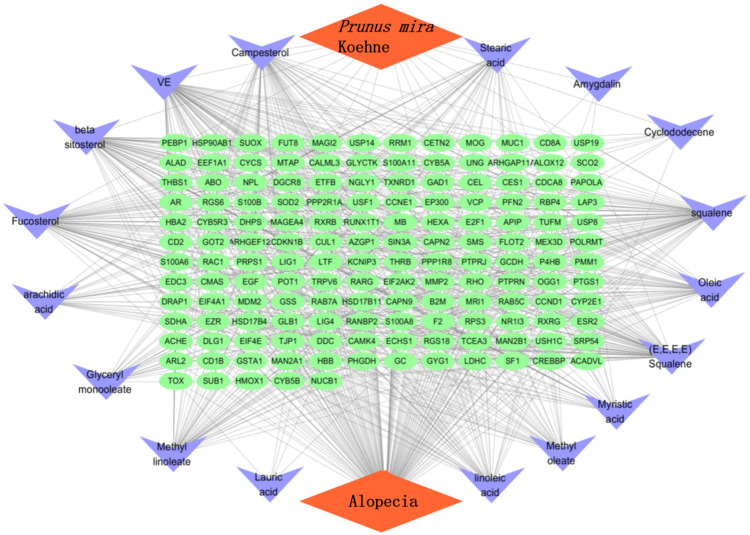
The kernel of *P. mira*-chemical composition-target-alopecia“ network analysis diagram.

**Figure 5 molecules-27-05242-f005:**
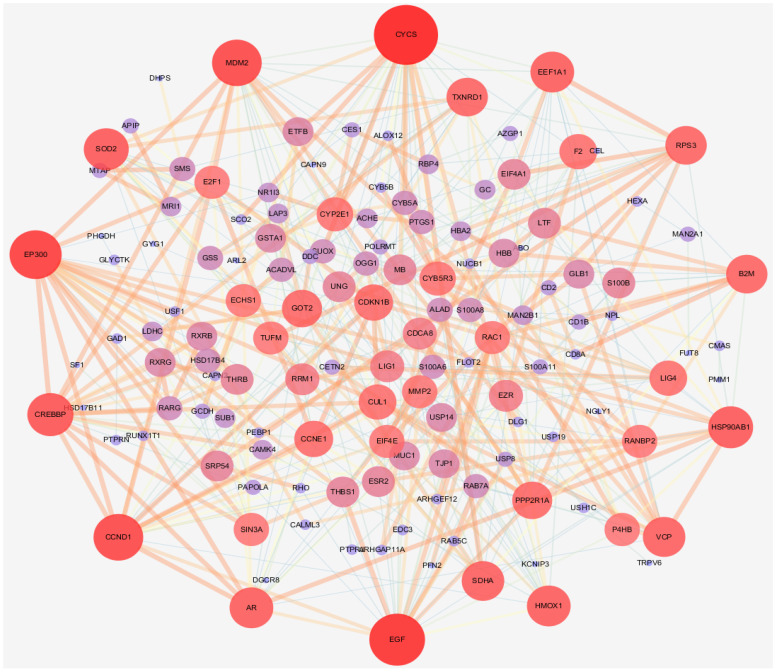
protein–protein interaction diagram.

**Figure 6 molecules-27-05242-f006:**
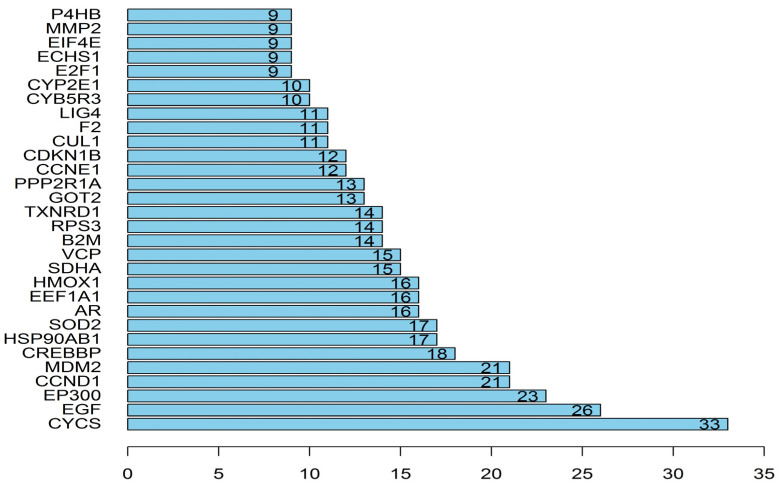
Degrees of freedom of the target.

**Figure 7 molecules-27-05242-f007:**
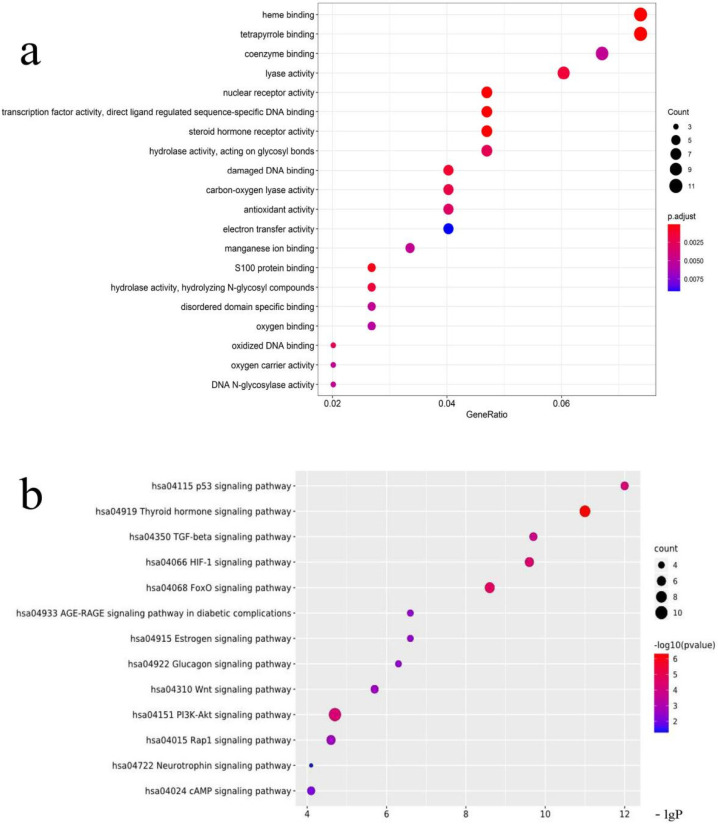
(**a**) GO analysis diagram. (**b**) KEGG enrichment map.

**Figure 8 molecules-27-05242-f008:**
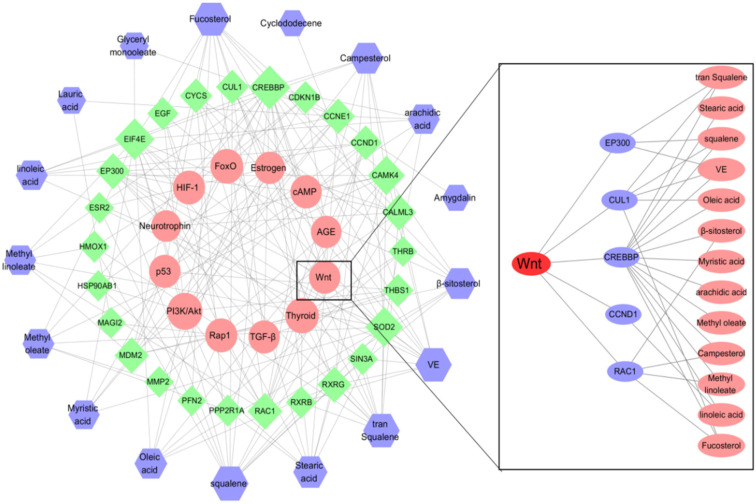
“The kernel of *P. mira* composition-target-signal pathway” network diagram.

**Figure 9 molecules-27-05242-f009:**
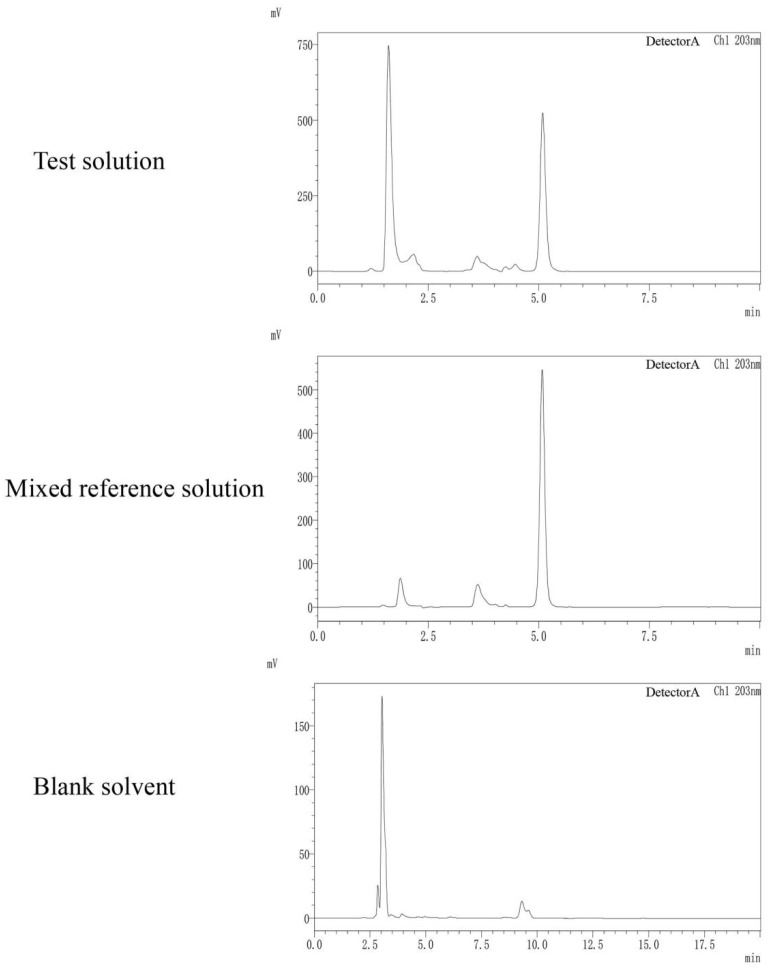
HPLC chromatograms of test solution, mixed reference solution and blank solvent.

**Figure 10 molecules-27-05242-f010:**
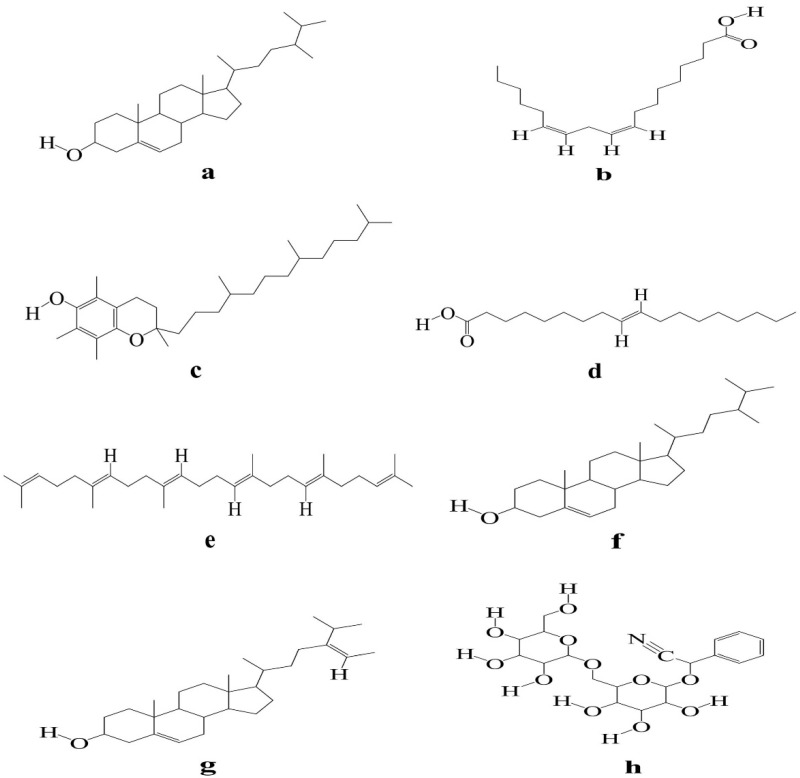
Structure diagram of eight chemical components (Note: (**a**) β-sitosterol, (**b**) linoleic acid, (**c**) vitamin E, (**d**) oleic acid, (**e**) trans-squalene, (**f**) campesterol, (**g**) fucosterol, (**h**) amygdalin).

**Figure 11 molecules-27-05242-f011:**
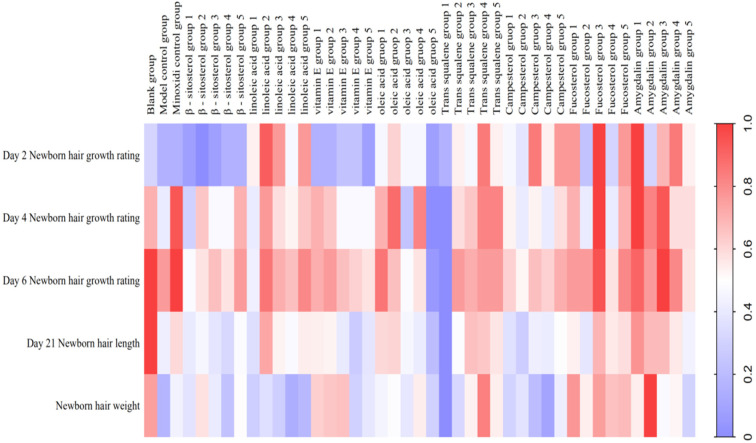
Heat map of the degree of influence of components in the kernel of *P. mira* on KM mice after depilation induced by sodium sulfide.

**Figure 12 molecules-27-05242-f012:**
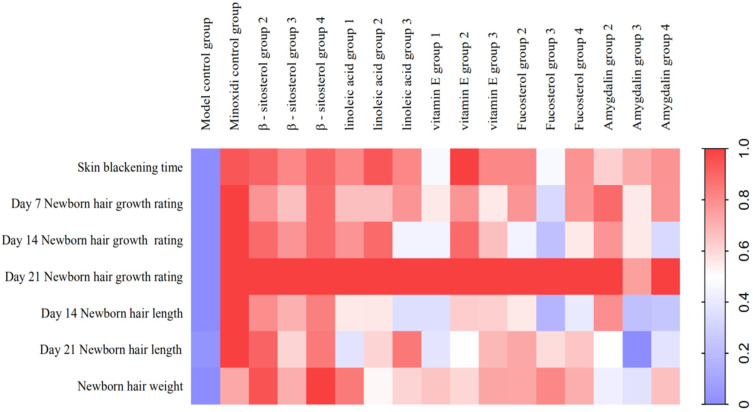
Heat map of the degree of influence of components of the kernel of *P. mira* on the depilation of C57BL/6 mice induced by sodium sulfide (note: the closer the color is too red, the more obvious the promoting effect of the test substance on hair growth).

**Figure 13 molecules-27-05242-f013:**
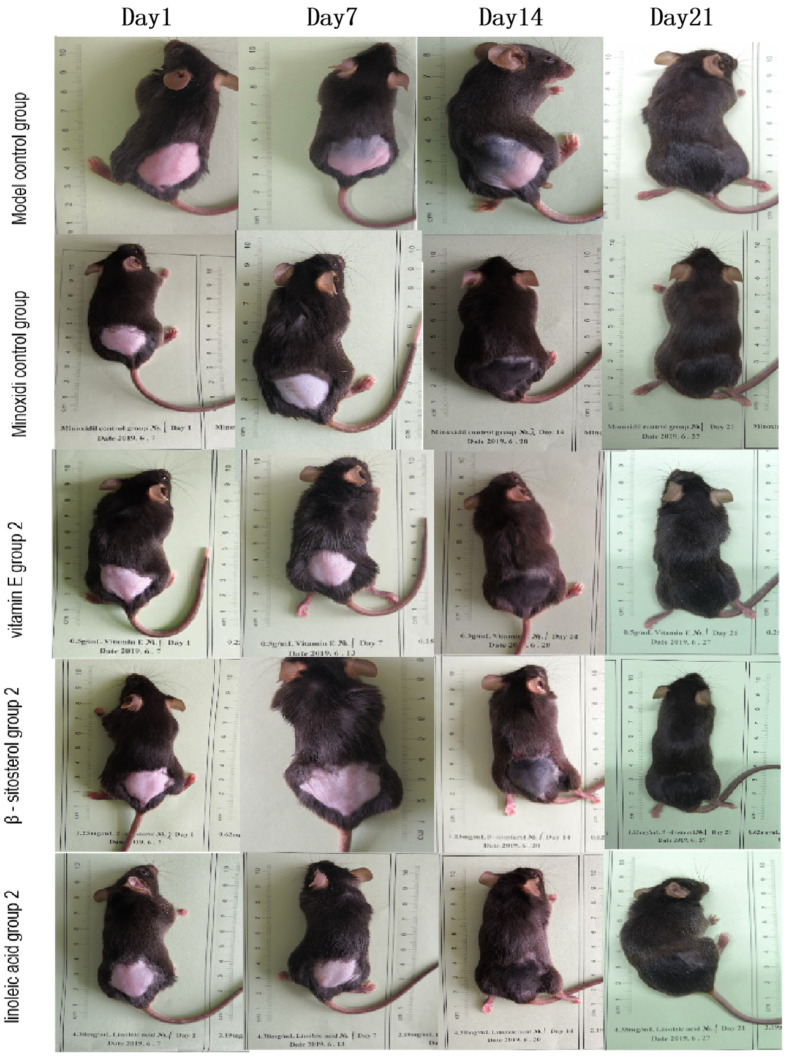
The effect of three effective components of the kernel of *P. mira* on the hair growth of C57BL/6 mice depilation model induced by sodium sulfide.

**Figure 14 molecules-27-05242-f014:**
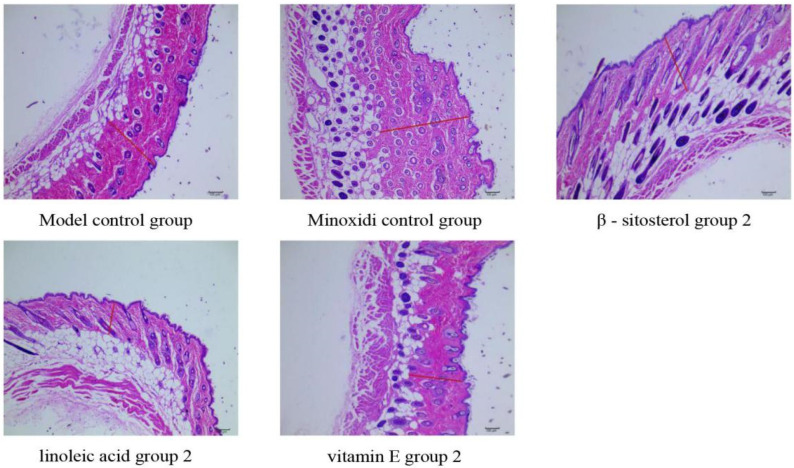
Number of hair follicles and dermal thickness (×100).

**Figure 15 molecules-27-05242-f015:**
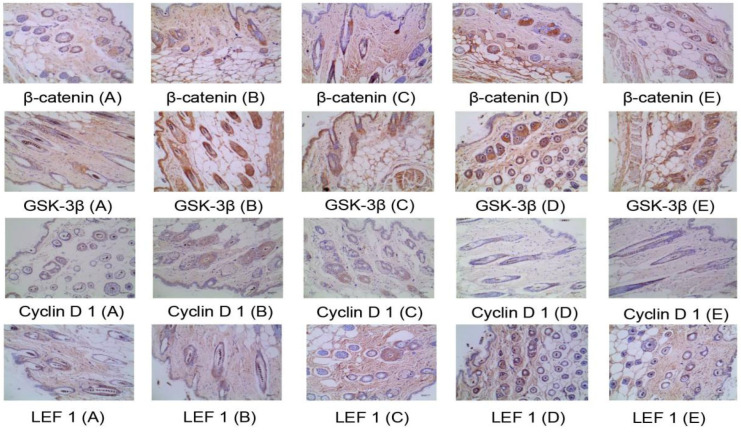
Immunohistochemistry diagram (×400). (**A**) Model control group; (**B**). Positive control group; (**C**). β-sitosterol group 2; (**D**). Linoleic acid E group 2; (**E**). Vitamin E group 2.

**Figure 16 molecules-27-05242-f016:**
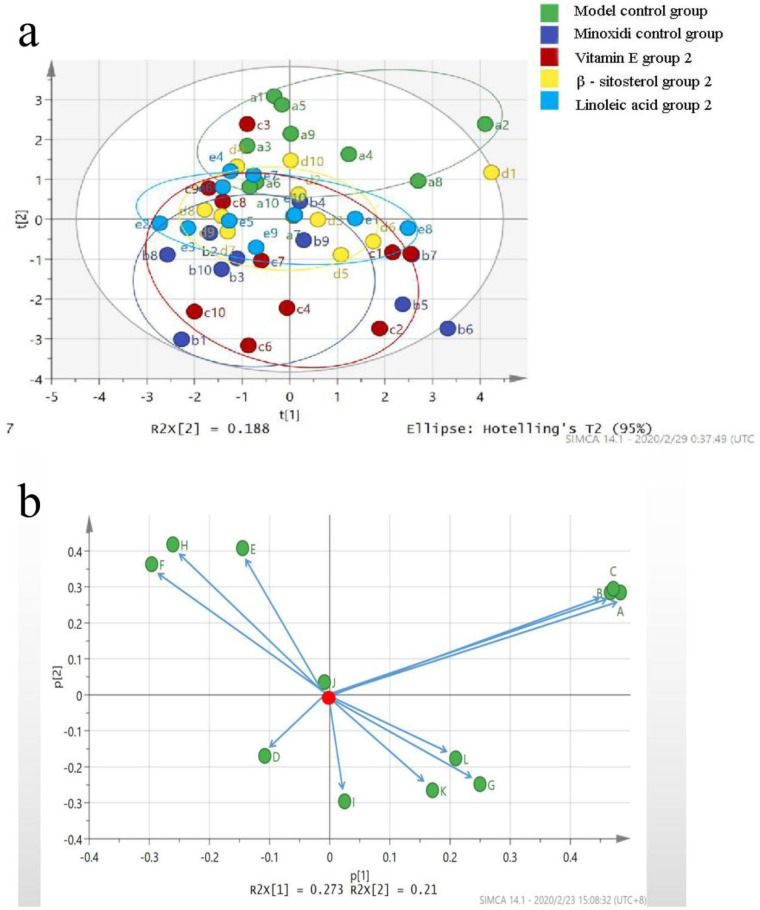
(**a**) PCA-X score diagram. (**b**): Contribution rates of indicators (a total number of hair follicles; B: number of primary hair follicles; C: number of secondary hair follicles; D: dermal thickness; E: mRNA expression of β-catenin; F: mRNA expression of GSK-3β; G: Cyclin D 1 mRNA expression; H: mRNA expression of LEF 1; I: protein expression of β-catenin; J: protein expression of GSK-3β; K: protein expression of Cyclin D 1; L: protein expression of LEF 1).

**Figure 17 molecules-27-05242-f017:**
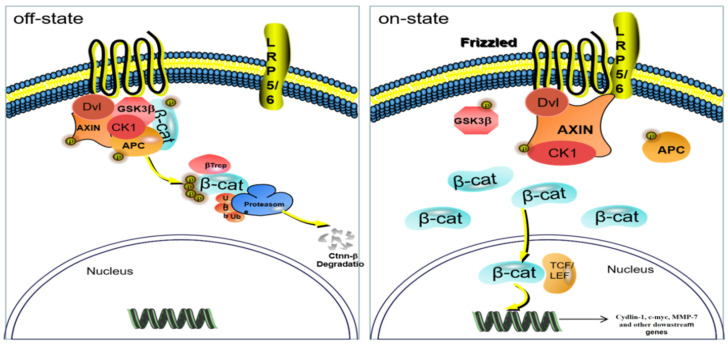
Wnt/β-catenin signaling pathway.

**Table 1 molecules-27-05242-t001:** Databases and analysis software related to Network pharmacology.

Tools	Function	Website
Pubchem	Compound structure search	https://pubchem.ncbi.nlm.nih.gov (accessed on 8 May 2020)
Pharmmapper	Compound target prediction	http://www.lilab-ecust.cn/Pharmmapper (accessed on 8 May 2020)
UniProt	Protein name correction	https://www.Uniprot.org (accessed on 10 May 2020)
Genecards	Disease target prediction	http://www.genecards.org (accessed on 13 May 2020)
String	Construct protein interaction map (PPI)	https://String—db.org (accessed on 23 May 2020)
Venny 2.1	Drawing	https://bioinfogp.cnb.csic.es/tools/Venny/ (accessed on 17 May 2020)
Metascape	Analyze the database	https://metascape.org/gp/index.html (accessed on 25 May 2020)
Cytospase3.7.1	Drawing software	——

**Table 2 molecules-27-05242-t002:** Drug group and dosage table of KM mice.

Group	Dose (mg/cm^2^/d)	Dose Volume (mL/cm^2^/d)	Drug Concentration (mg/mL)	Group	Dose (mg/cm^2^/d)	Dose Volume (mL/cm^2^/d)	Drug Concentration (mg/mL)
Blank group	—	0.025	—	Model control group	—	0.025	—
Minoxidi control group	0.500	0.025	20.00	—	—	—	—
β-sitosterol group 1	0.123	0.025	4.90	linoleic acid group 1	0.313	0.025	12.50
β-sitosterol group 2	0.061	0.025	2.45	linoleic acid group 2	0.156	0.025	6.25
β-sitosterol group 3	0.031	0.025	1.23	linoleic acid group 3	0.078	0.025	3.13
β-sitosterol group 4	0.016	0.025	0.62	linoleic acid group 4	0.039	0.025	1.57
β-sitosterol group 5	0.008	0.025	0.31	linoleic acid group 5	0.020	0.025	0.79
vitamin E group 1	6.250	0.025	250.00	oleic acid group 1	0.00055	0.025	0.02200
vitamin E group 2	3.125	0.025	125.00	oleic acid group 2	0.00028	0.025	0.01100
vitamin E group 3	1.563	0.025	62.50	oleic acid group 3	0.00015	0.025	0.00600
vitamin E group 4	0.781	0.025	31.25	oleic acid group 4	0.00008	0.025	0.00300
vitamin E group 5	0.391	0.025	15.63	oleic acid group 5	0.00004	0.025	0.00150
Trans squalene group 1	0.123	0.025	4.90	Campesterol group 1	0.060	0.025	2.39
Trans squalene group 2	0.061	0.025	2.45	Campesterol group 2	0.030	0.025	1.20
Trans squalene group 3	0.031	0.025	1.23	Campesterol group 3	0.015	0.025	0.60
Trans squalene group 4	0.016	0.025	0.62	Campesterol group 4	0.008	0.025	0.30
Trans squalene group 5	0.008	0.025	0.31	Campesterol group 5	0.004	0.025	0.15
Fucosterol group 1	0.123	0.025	4.90	Amygdalin group 1	0.123	0.025	4.90
Fucosterol group 2	0.061	0.025	2.45	Amygdalin group 2	0.061	0.025	2.45
Fucosterol group 3	0.031	0.025	1.23	Amygdalin group 3	0.031	0.025	1.23
Fucosterol group 4	0.016	0.025	0.62	Amygdalin group 4	0.016	0.025	0.62
Fucosterol group 5	0.008	0.025	0.31	Amygdalin group 5	0.008	0.025	0.31

**Table 3 molecules-27-05242-t003:** Rating standard table of KM mice and C57BL/6 mice newborn hair condition.

Rating	Standard (KM Mice)	Standard (C57BL/6 Mice)
I	Hairless growth	The skin of the administration area is pink
II	Shallow hair overgrown depilated area	The skin in the depilatory area is gray
III	The length and density of new hair is about one-half of the unhaired area	The skin in the epilation area is black
IV	No difference between newborn hair growth and unhaired areas	Hair grows in the depilatory area

**Table 4 molecules-27-05242-t004:** Genomic DNA removal reaction system.

Reagent	Volume (μL)
5 × gDNA Eraser Buffer	2
gDNA Eraser	1
Total RNA	2
RNase Free dH_2_O	5
Total	10

Note: After adding each reagent in sequence, 42 °C, 2 min.

**Table 5 molecules-27-05242-t005:** Reverse transcription reaction system.

Reagent	Volume (μL)
5 × PrimeScript Buffer 2	4
PrimeScript RT Enzyme Mix I	1
RT Primer Mix	1
RNA	10
RNase Free dH_2_O	4
Total	20

Note: Add the reagents one by one and put them on the PCR machine for reaction.

**Table 6 molecules-27-05242-t006:** Primers and base sequences used in detection.

Primer Name	Upstream	Downstream	Citation Length, bp
β-actin	gaagatcaagatcattgctcc	tactcctgcttgctgatcca	111
Cyclin D 1	ccagaggcggatgagaacaagcagac	tgtgcggtagcaggagaggaagttgt	183
GSK3β	acagtggtgtggatcagttggtggaa	ccagaggcggatgagaacaagcagac	153
LEF 1	caacgggcatgaggtggtcagacaag	agtgctcgtcgctgtaggtgatgagg	293
β-actin	gaagatcaagatcattgctcc	tactcctgcttgctgatcca	296

**Table 7 molecules-27-05242-t007:** PCR reaction system.

Reagent	Volume (μL)
2 × Real PCR EasyTM Mix-SYBR	10.0
Forward Primer (10 μM)	0.8
Reverse Primer (10 μM)	0.8
Template (DNA)	2.0
ddH_2_O	6.4
Total	20.0

**Table 8 molecules-27-05242-t008:** PCR reaction program.

Reaction Temperature	Time	Remarks
95 °C	30 s	Predenaturation
95 °C	5 s	transsexual
55 °C	30 s	annealing
72 °C	30 s	Fully extend to collect fluorescence

Note: 45 cycles.

**Table 9 molecules-27-05242-t009:** Chemical constituents of the kernel of *P. mira*.

Compound	CID	Compound	CID
amygdalin	656516	arachidic acid	10467
beta-sitosterol	222284	campesterol	173183
cyclododecene	637538	fucosterol	5281328
glyceryl monooleatemonooleate	5283468	methyl linoleate	5284421
methyl oleate	5364509	myristic acid	11005
lauric acid	3893	linoleic acid	5280450
oleic acid	445639	squalene	638072
stearic acid	5281	*trans*-squalene	638072
vitamin E	14985		

**Table 10 molecules-27-05242-t010:** Potential targets of the kernel of *P. mira* in the treatment of alopecia.

Gene Name	Gene Name	Gene Name	Gene Name	Gene Name	Gene Name
*ABO*	*ACADVL*	*ACHE*	*ALAD*	*ALOX12*	*APIP*
*AR*	*ARHGAP11*	*ARHGEF12*	*ARL2*	*AZGP1*	*B2M*
*CALmL3*	*CAMK4*	*CAPN2*	*CAPN9*	*CCND1*	*CCNE1*
*CD1B*	*CD2*	*CD8A*	*CDCA8*	*CDKN1B*	*CEL*
*CES1*	*CETN2*	*CMAS*	*CREBBP*	*CUL1*	*CYB5A*
*CYB5B*	*CYB5R3*	*CYCS*	*CYP2E1*	*DDC*	*DGCR8*
*DHPS*	*DLG1*	*DRAP1*	*E2F1*	*ECHS1*	*EDC3*
*EEF1A1*	*EGF*	*EIF2AK2*	*EIF4A1*	*EIF4E*	*EP300*
*ESR2*	*ETFB*	*EZR*	*F2*	*FLOT2*	*FUT8*
*GAD1*	*GC*	*GCDH*	*GLB1*	*GLYCTK*	*GOT2*
*GSS*	*GSTA1*	*GYG1*	*HBA2*	*HBB*	*HEXA*
*HMOX1*	*HSD17B11*	*HSD17B4*	*HSP90AB1*	*KCNIP3*	*LAP3*
*LDHC*	*LIG1*	*LIG4*	*LTF*	*MAGEA4*	*MAGI2*
*MAN2A1*	*MAN2B1*	*MB*	*MDM2*	*MEX3D*	*MMP2*
*MOG*	*MRI1*	*MTAP*	*MUC1*	*NGLY1*	*NPL*
*NR1I3*	*NUCB1*	*OGG1*	*P4HB*	*PAPOLA*	*PEBP1*
*PFN2*	*PHGDH*	*PMM1*	*POLRMT*	*POT1*	*PPP1R8*
*PPP2R1A*	*PRPS1*	*PTGS1*	*PTPRJ*	*PTPRN*	*RAB5C*
*RAB7A*	*RAC1*	*RANBP2*	*RARG*	*RBP4*	*RGS18*
*RGS6*	*RHO*	*RPS3*	*RRM1*	*RUNX1T1*	*RXRB*
*RXRG*	*S100A11*	*S100A6*	*S100A8*	*S100B*	*SCO2*
*SDHA*	*SF1*	*SIN3A*	*SMS*	*SOD2*	*SRP54*
*SUB1*	*SUOX*	*TCEA3*	*THBS1*	*THRB*	*TJP1*
*TOX*	*TRPV6*	*TUFM*	*TXNRD1*	*UNG*	*USF1*
*USH1C*	*USP14*	*USP19*	*USP8*	*VCP*	—

**Table 11 molecules-27-05242-t011:** Correlation between standard concentration and area under peak of chemical components in the kernel of *P. mira*.

Chemical Component	Linear Regression Equation	Correlation Coefficient (R)	Linear Range (µg/mL)	Precision	Repeatability	Stability	Average Value of Sample Recovery	Sample Recovery Rate
Amygdalin	y = 10749x + 1018982	0.9993	80.00–1280.00	2.60%	2.40%	2.8%	95.04%	1.60%

**Table 12 molecules-27-05242-t012:** Contents of amygdalin in the kernels of *P. mira*.

Origin	Altitude (m)	Longitude	Latitude	Oil Yield (%)	Content (mg/g)
					Amygdalin
Xulong Township, Derong County	2935	99°13′7005″	28°74′1808″	38.08	14.8 ± 0.2
Zhongza Town, Batang County	2929	99°19′1089″	29°21′2186″	37.89	11.0 ± 1.8
Bajiaolou Township, Yajiang County	2719	101°06′1872″	30°06′0709″	37.94	15.1 ± 0.3
Chitu Township, Daocheng County	3164	100°16′1899″	28°37′3726″	38.13	16.2 ± 0.3
Malkang Forestry Bureau	2630	102°13′4203″	31°55′2159″	38.11	13.9 ± 0.7
Xinlong County	3066	100°31′1368″	30°93′9169″	38.04	25.9 ± 0.7
Zhengdou Township, Xiangcheng County	2750	99°31′2148″	29°05′4042″	37.53	11.1 ± 2.7
Wachang Town, Muli County	2577	100°50′2772″	28°09′4138″	38.23	15.1 ± 4.8
Pusharong Township, Kangding City	2922	101°19′1446″	29°32′1447″	38.05	19.6 ± 0.1
Jiaer Town, Jiulong County	2823	101°30′3892″	28°59′1512″	38.04	21.2 ± 0.3
Nixi Township, Shangri-La City *	3135	99°50′6456″	28°04′7398″	37.87	12.1 ± 1.8
Benzilan Town, Deqin County *	2220	99°16′4288″	28°14′2346″	37.94	13.5 ± 2.7

Note: Those marked with * are areas under the jurisdiction of Yunnan Province, China, and the others are areas under the jurisdiction of Sichuan Province, China; *n* = 3.

**Table 13 molecules-27-05242-t013:** Newborn hair growth rating, the newborn hair length and hair weight in KM mice (*n* = 10).

Group	Dose (mg/cm^2^/d)	Newborn Hair Growth Rating															Newborn Hair Length (mm)	Newborn Hair Weight (mg)
		Day 2					Day 4					Day 6						
		I	II	III	IV	*p*	I	II	III	IV	*p*	I	II	III	IV	*p*		
Blank group	—	5	5	0	0	0.660	0	3	7	0	0.024 *	0	0	3	7	0.129 *	6.72 ± 0.53 **	2.82 ± 1.26 **
Model control group	—	7	3	0	0	—	0	8	2	0	—	0	2	4	4	—	3.73 ± 1.7	1.33 ± 0.59
Minoxidi control group	0.500	7	3	0	0	1.000	0	2	5	3	0.003 **	0	0	3	7	0.129	4.63 ± 0.85	2.03 ± 0.51 *
β-sitosterol group 1	0.123	8	2	0	0	0.016 *	3	4	3	0	0.113	1	3	5	1	0.129	3.64 ± 1.70	1.86 ± 0.74
β-sitosterol group 2	0.061	9	1	0	0	0.001 *	0	4	6	0	0.044 *	0	4	4	2	0.227	3.91 ± 1.95	2.38 ± 1.27 *
β-sitosterol group 3	0.031	8	2	0	0	0.016 *	0	7	3	0	0.628 *	0	2	6	2	0.262	3.51 ± 0.55	1.95 ± 0.98
β-sitosterol group 4	0.016	7	3	0	0	1.000	0	7	3	0	0.628 *	0	3	6	1	0.227	3.23 ± 1.42	1.49 ± 0.75
β-sitosterol group 5	0.008	7	3	0	0	1.000	1	2	6	1	0.063 *	1	1	3	5	0.815	4.02 ± 1.83	2.20 ± 1.36
linoleic acid group 1	0.313	2	8	0	0	0.064	1	6	3	0	0.928	2	3	3	2	0.137	3.35 ± 2.69	1.64 ± 1.42
linoleic acid group 2	0.156	0	7	3	0	0.021 *	0	2	8	0	0.005 **	0	2	2	6	0.525	5.33 ± 1.69	1.82 ± 1.42
linoleic acid group 3	0.078	1	7	2	0	0.003 **	1	3	6	0	0.177	1	1	4	4	0.380	4.28 ± 2.11	1.64 ± 1.11
linoleic acid group 4	0.039	3	7	0	0	0.081	1	4	5	0	0.350	1	2	3	4	0.758	3.97 ± 12.33	1.26 ± 0.71
linoleic acid group 5	0.020	2	5	3	0	0.181	1	2	7	0	0.060	1	2	0	7	0.137	4.36 ± 2.41	1.38 ± 0.81
vitamin E group 1	6.250	7	3	0	0	1.000	0	3	7	0	0.024 *	0	2	5	3	0.754	4.33 ± 2.03	2.50 ± 1.21 *
vitamin E group 2	3.125	7	3	0	0	1.000	0	4	6	0	0.044 *	0	2	4	4	1.000	4.28 ± 1.66	2.56 ± 0.97 **
vitamin E group 3	1.563	6	4	0	0	0.673	0	7	3	0	0.628	0	3	4	3	0.596	3.62 ± 1.55	2.59 ± 1.03 **
vitamin E group 4	0.781	6	4	0	0	0.673	0	7	3	0	0.628	0	4	3	3	0.475	2.92 ± 2.38	1.66 ± 1.02
vitamin E group 5	0.391	8	2	0	0	0.660	2	4	3	1	0.805	1	4	1	4	0.439	3.52 ± 2.12	1.90 ± 1.33
oleic acid group 1	0.00055	3	7	0	0	0.081	2	1	5	2	0.132	2	0	0	8	0.240	4.65 ± 2.34	2.05 ± 1.08
oleic acid group 2	0.00028	1	9	0	0	0.058	1	1	5	3	0.051	1	1	5	3	0.686	4.70 ± 2.39	2.14 ± 1.32
oleic acid group 3	0.00015	3	7	0	0	0.081	4	3	3	0	0.375	2	4	0	4	0.279	3.90 ± 2.20	1.91 ± 1.35
oleic acid group 4	0.00008	3	7	0	0	0.081	1	2	4	3	0.054	3	1	1	5	0.700	3.67 ± 2.25	2.30 ± 1.32
oleic acid group 5	0.00004	8	2	0	0	0.897	7	1	2	0	0.022 *	5	3	2	0	0.001 **	2.58 ± 2.36	1.66 ± 1.55
Tran squalene group 1	0.123	7	3	0	0	1.000	6	3	1	0	0.013 *	6	2	2	0	0.002 **	1.52 ± 1.73 **	0.90 ± 1.10 *
Trans squalene group 2	0.061	2	8	0	0	0.064	1	5	2	2	0.493	0	2	4	4	1.000	4.03 ± 2.06	1.72 ± 0.77
Trans squalene group 3	0.031	3	7	0	0	0.081	2	2	4	2	0.267	2	0	3	5	0.876	4.97 ± 2.34	2.27 ± 1.26
Trans squalene group 4	0.016	1	6	3	0	0.840	1	2	4	3	0.054	1	2	1	6	0.753	4.88 ± 1.44	3.03 ± 1.85
Trans squalene group 5	0.008	2	8	0	0	0.064	0	4	3	3	0.050	0	2	4	4	1.000	4.47 ± 1.38	2.29 ± 1.54
Campesterol group 1	0.060	3	7	0	0	0.081	2	2	6	0	0.330	2	2	1	5	0.758	3.40 ± 2.75	1.68 ± 1.59
Campesterol group 2	0.030	4	6	0	0	0.196	3	2	5	0	0.812	3	0	4	3	0.438	2.96 ± 2.48	1.86 ± 0.77
Campesterol group 3	0.015	1	6	3	0	0.840	1	5	3	1	0.546	1	2	3	4	0.758	3.72 ± 2.45	1.42 ± 0.86
Campesterol group 4	0.008	3	6	1	0	0.840	2	4	4	0	0.868	2	2	1	5	0.758	3.68 ± 2.89	1.15 ± 0.97
Campesterol group 5	0.004	1	7	2	0	0.053	1	3	6	0	0.177	1	2	2	5	1.000	4.14 ± 2.04	1.97 ± 0.97
Fucosterol group 1	0.123	2	5	3	0	0.053	2	2	3	3	0.237	1	1	3	5	0.815	4.32 ± 2.20	2.88 ± 1.86
Fucosterol group 2	0.061	7	2	1	0	0.892	2	4	4	0	0.868	2	0	2	6	0.049 *	3.53 ± 2.38	2.29 ± 1.53
Fucosterol group 3	0.031	1	4	5	0	0.011	0	2	4	4	0.002 **	0	0	4	6	0.037 *	5.14 ± 1.21 *	2.83 ± 0.99
Fucosterol group 4	0.016	5	5	0	0	0.412	3	2	5	0	0.812	1	3	3	3	0.395	4.43 ± 1.56	2.59 ± 1.80
Fucosterol group 5	0.008	2	5	3	0	0.053	1	3	4	2	0.138	0	3	1	6	0.693	4.58 ± 1.70	2.64 ± 1.21
Amygdalin group 1	0.123	0	6	4	0	0.055	0	2	4	4	0.014 *	0	2	1	7	0.034 *	5.58 ± 1.63 *	2.29 ± 0.74
Amygdalin group 2	0.061	7	1	2	0	0.384	1	2	4	3	0.054	1	1	3	5	0.815	5.05 ± 2.00	3.46 ± 1.80
Amygdalin group 3	0.031	2	6	2	0	0.376	0	3	3	4	0.031 *	0	0	3	7	0.029 *	5.04 ± 1.41	2.11 ± 0.97
Amygdalin group 4	0.016	0	8	2	0	0.064	0	5	5	0	0.178	0	2	3	5	0.754	4.43 ± 1.56	2.24 ± 1.28
Amygdalin group 5	0.008	4	4	2	0	0.129	2	2	5	1	0.298	2	1	4	3	0.486	3.78 ± 2.16	1.67 ± 1.23

Note: Compared with the model control group, * *p* ≤ 0.05, ** *p* ≤ 0.01.

**Table 14 molecules-27-05242-t014:** Effects of five effective components in the kernel of *P. mira* on the newborn hair growth rating of C57BL/6 mice depilation model induced by sodium sulfide.

Group	Dose (mg/cm^2^/d)	Skin Blackening Time (Day)	Newborn Hair Growth Rating														
			Day 7					Day 14					Day 21				
			I	II	III	IV	*p*	I	II	III	IV	*p*	I	II	III	IV	*p*
Model control group	—	11.70 ± 3.64	8	2	0	0	—	1	1	4	4	—	0	1	2	7	—
Minoxidi control group	0.500	8.70 ± 1.70 *	0	9	1	0	*p* ≤ 0.01	0	0	0	10	*p* ≤ 0.05	0	0	0	10	*p* ≥ 0.05
β-sitosterol group 2	0.061	8.80 ± 2.04 *	3	5	2	0	*p* ≤ 0.01	0	0	1	9	*p* ≤ 0.05	0	0	0	10	*p* ≥ 0.05
β-sitosterol group 3	0.031	9.10 ± 1.73	2	8	0	0	*p* ≤ 0.01	0	0	2	8	*p* ≥ 0.05	0	0	0	10	*p* ≥ 0.05
β-sitosterol group 4	0.016	8.80 ± 1.03 *	1	8	1	0	*p* ≤ 0.01	0	0	1	9	*p* ≤ 0.05	0	0	0	10	*p* ≥ 0.05
linoleic acid group 1	0.313	9.10 ± 1.52	3	6	1	0	*p* ≤ 0.01	0	0	2	8	*p* ≥ 0.05	0	0	0	10	*p* ≥ 0.05
linoleic acid group 2	0.156	8.70 ± 0.82 *	2	8	0	0	*p* ≤ 0.01	0	0	1	9	*p* ≤ 0.05	0	0	0	10	*p* ≥ 0.05
linoleic acid group 3	0.078	9.10 ± 1.29	2	7	1	0	*p* ≤ 0.01	0	1	3	6	*p* ≥ 0.05	0	0	0	10	*p* ≥ 0.05
vitamin E group 1	0.313	10.20 ± 2.62	3	7	0	0	*p* ≤ 0.01	0	2	1	7	*p* ≥ 0.05	0	0	0	10	*p* ≥ 0.05
vitamin E group 2	0.156	8.50 ± 1.18 *	2	7	1	0	*p* ≤ 0.01	0	0	1	9	*p* ≤ 0.05	0	0	0	10	*p* ≥ 0.05
vitamin E group 3	0.078	9.10 ± 1.79	3	7	0	0	*p* ≤ 0.01	0	1	1	8	*p* ≥ 0.05	0	0	0	10	*p* ≥ 0.05
Fucosterol group 2	0.061	9.10 ± 1.10 *	1	9	0	0	*p* ≤ 0.01	0	1	3	6	*p* ≥ 0.05	0	0	0	10	*p* ≥ 0.05
Fucosterol group 3	0.031	10.20 ± 2.62	5	5	0	0	*p* ≥ 0.05	1	0	4	5	*p* ≥ 0.05	0	0	0	10	*p* ≥ 0.05
Fucosterol group 4	0.016	9.20 ± 2.49	2	7	1	0	*p* ≤ 0.01	0	1	2	7	*p* ≥ 0.05	0	0	0	10	*p* ≥ 0.05
Amygdalin group 2	0.061	9.70 ± 1.83	0	10	0	0	*p* ≤ 0.01	0	0	2	8	*p* ≥ 0.05	0	0	0	10	*p* ≥ 0.05
Amygdalin group 3	0.031	9.40 ± 1.43	3	7	0	0	*p* ≤ 0.01	0	1	2	7	*p* ≥ 0.05	0	0	1	9	*p* ≥ 0.05
Amygdalin group 4	0.016	9.20 ± 0.92	2	7	1	0	*p* ≤ 0.01	0	1	4	5	*p* ≥ 0.05	0	0	0	10	*p* ≥ 0.05

Note: Compared with the model control group, * *p* ≤ 0.05, ** *p* ≤ 0.01.

**Table 15 molecules-27-05242-t015:** Effects of five active ingredients in the kernel of *P. mira* on the newborn hair length and weight of C57BL/6 mice depilation model induced by sodium sulfide.

Group	Dose (mg/cm^2^/d)	Number of Animals (Number)	Newborn Hair Length (Day 7, cm)	Newborn Hair Length (Day 14, cm)	Newborn Hair Weight (g)
Model control group	—	10	0.138 ± 0.151	0.625 ± 0.186	0.00154 ± 0.00071
Minoxidi control group	0.500	10	0.420 ± 0.133 **	0.816 ± 0.161 **	0.00260 ± 0.00035 **
β-sitosterol group 2	0.061	10	0.363 ± 0.203 **	0.798 ± 0.116 **	0.00284 ± 0.00044 **
β-sitosterol group 3	0.031	10	0.336 ± 0.153 **	0.740 ± 0.107	0.00258 ± 0.00037 **
β-sitosterol group 4	0.016	10	0.373 ± 0.113 **	0.786 ± 0.120 **	0.00299 ± 0.00051 **
linoleic acid group 1	0.313	10	0.296 ± 0.124 **	0.693 ± 0.071	0.00277 ± 0.00037 **
linoleic acid group 2	0.156	10	0.297 ± 0.152 **	0.739 ± 0.152	0.00230 ± 0.00050 **
linoleic acid group 3	0.078	10	0.238 ± 0.188	0.787 ± 0.138 **	0.00243 ± 0.00053 **
vitamin E group 1	0.313	10	0.238 ± 0.173	0.693 ± 0.156	0.00249 ± 0.00029 **
vitamin E group 2	0.156	10	0.316 ± 0.167 **	0.716 ± 0.131	0.00242 ± 0.00028 **
vitamin E group 3	0.078	10	0.314 ± 0.064 **	0.755 ± 0.065	0.00261 ± 0.00063 **
Fucosterol group 2	0.061	10	0.296 ± 0.145 **	0.762 ± 0.131	0.00260 ± 0.00050 **
Fucosterol group 3	0.031	10	0.186 ± 0.173	0.735 ± 0.129	0.00272 ± 0.00056 **
Fucosterol group 4	0.016	10	0.252 ± 0.155	0.747 ± 0.087	0.00257 ± 0.00041 **
Amygdalin group 2	0.061	10	0.362 ± 0.101 **	0.717 ± 0.093	0.00216 ± 0.00041 **
Amygdalin group 3	0.031	10	0.201 ± 0.146	0.619 ± 0.182	0.00208 ± 0.00056 **
Amygdalin group 4	0.016	10	0.208 ± 0.121	0.693 ± 0.100	0.00250 ± 0.00058

Note: Compared with the model control group, * *p* ≤ 0.05, ** *p* ≤ 0.01.

**Table 16 molecules-27-05242-t016:** Effects of three effective components of the kernel of *P. mira* on the number of hair follicles and dermal thickness in the experiment of C57BL/6 mice depilation model induced by sodium sulfide.

Group	Dose (mg/cm^2^/d)	Dermis Thickness (nm)	Dermis Thickness (Number)	Secondary Hair Follicle (Number)	Total Number of Hair Follicles (Number)
Model control group	—	360.15 ± 150.44	22.17 ±12.25	36.33 ± 21.02	59.7 ± 29.26
Minoxidi control group	0.50	344.63 ± 137.09	20.10 ± 12.98	43.37 ± 31.69	63.47 ± 44.28
β-sitosterol group 2	0.061	345.24 ± 107.93	19.30 ± 11.15	41.17 ± 27.59	60.47 ± 38.15
linoleic acid group 2	0.156	381.06 ± 156.68	14.27 ± 10.18	24.83 ± 21.15	39.10 ± 32.96
vitamin E group 2	0.156	350.47 ± 150.87	20.63 ± 11.85	37.73 ± 22.60	58.37 ± 34.36

**Table 17 molecules-27-05242-t017:** Effects of three effective components of the kernel of *P. mira* on the mRNA expression of four targets in the Wnt/β-catenin signaling pathway in the skin tissue of C57BL/6 mice ($\bar{x}\pm s$ ).

Group	Dose (mg/cm^2^/d)	β-catenin	GSK3β	Cyclin D 1	LEF 1
Model control group	—	0.23 ± 0.11	0.36 ± 0.16	1.61 ± 0.68	0.43 ± 0.17
Minoxidi control group	0.50	0.71 ± 0.27 **	0.79 ± 0.27 **	1.07 ± 0.21 *	0.81 ± 0.20 **
β-sitosterol group 2	0.061	0.27 ± 0.26	0.57 ± 0.18 **	1.31 ± 0.28	0.61 ± 0.18 *
linoleic acid group 2	0.156	0.39 ± 0.26	0.72 ± 0.26 **	1.41 ± 0.26	0.70 ± 0.32 *
vitamin E group 2	0.156	0.50 ± 0.50	0.73 ± 0.19 **	1.21 ± 0.43	0.91 ± 0.49 **

Note: Compared with the model control group, * *p* ≤ 0.05, ** *p* ≤ 0.01.

**Table 18 molecules-27-05242-t018:** Effects of three effective components of the kernel of *P. mira* on the protein expression of four targets in the Wnt/β-catenin signaling pathway in the skin tissue of C57BL/6 mice ($\bar{x}\pm s$ ).

Group	Dose (mg/cm^2^/d)	Number of Animals (Number)	β-catenin	GSK3β	Cyclin D 1	LEF 1
Model control group	—	10	0.222 ± 0.008	0.290 ± 0.018	0.195 ± 0.025	0.242 ± 0.010
Minoxidi control group	0.50	10	0.220 ± 0.010	0.278 ± 0.011	0.188 ± 0.028	0.241 ± 0.012
β-sitosterol group 2	0.061	10	0.218 ± 0.006	0.284 ± 0.022	0.183 ± 0.031	0.231 ± 0.015
linoleic acid group 2	0.156	10	0.217 ± 0.008	0.278 ± 0.009	0.171 ± 0.021	0.229 ± 0.016
vitamin E group 2	0.156	10	0.216 ± 0.006	0.279 ± 0.012	0.192 ± 0.030	0.232 ± 0.017

Note: Compared with the model control group, * *p* ≤ 0.05, ** *p* ≤ 0.01.

## Data Availability

Not applicable.

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
