# Peer review of "Identification of Hair Growth Promoting Components in the Kernels of Prunus mira Koehne and Their Mechanism of Action"

_molecules, 2022, doi:10.3390/molecules27165242_

Round 1

Reviewer 1 Report

1.      In this article, the authors complete comprehensive experiments to derive the possible MOA of P. mira for the treatment of alopecia.

2.      I would suggest the authors add the common structure for the active components in the study.

3.      Page 21, Line 450, 3.2.1 Materials. Did the authors verify the batch-to-batch consistency of the extracts from 12 batches? What are the acceptance criteria for the purity of the test components?

4.      If the potentially active components were already known, such as b-sitosterol, linoleic acid, amygdalin, etc. Did the authors use pure compounds instead of extracts from natural sources to evaluate the difference? Did the authors prove the benefits from natural sources outweigh the pure compounds?

Reviewer 2 Report

It is opinion of the reviewer that this paper before acceptance needs several corrections. My individual comments are listed below.

In “Molecules”, the paper parts are in order of Introduction, Results, Discussion, Material and Methods. The author must take in under consideration when correcting this paper.

The table numbering must be corrected.

The title – Capital first letters are needed.

L. 7-9 -  The authors’ initial as well as e-mail addresses must be added.

L. 61 – Remove “mineral”.

L. 63 – Full name of “Vc”.

L. 70 – Why the authors mentioned walnuts in this place?

L. 71 – α-tocopherol has activity of vitamin E.

L. 124-181 – The description of the network pharmacology is too long and should be reduced.

L. 190 – “Error! Reference source not found”?

L. 202, 524 – The vitamin E activity exhibit tocopherols and tocotrienols. It is not clear what “vitamin E” in this paper means. The composition of “vitamin E” (see l. ) must be reported.

L. 207 – The names of chemical compounds should be written with lower case letters (for example, “amygdalin, beta-sitosterol, …”.

L. 209, 312 – “trans” should be written with italic.

L. 268 – It should be “acetonitrile-water”.

A part 2.2.1 should be included in Material and Methods.

L. 281 – What does it means “The relationship of oleic acid, …”?

L. 283– What does it means “The methodological investigation RSD% …”?

The title of Table 9 is wrong – it was correlation between concentration of the standard and area under a peak.

Table 9 – Remove “(RSD)” from the table head.

Figure 9 – Remove 3 standards curves – The same results are reported in table 9.

Table 10 – It should be “Content of oleic acid, linoleic acid, and amygdalin … “. “n = 3” should be reported in footnote.

Table 10 - ± SD should be added.

Table 10 – The results should be reported with one digital after decimal point.

Figure 11 – The quality of this figure is very pure. The content was reported in table 10. This figure should be removed.

Table 14,  15 – Remove “Number of animal(number) from the table. It should be reported only in Material and Methods section.

Table 14 – What does it means “(number)”? Correction is needed.

L. 451 – How big were batches of the kernel?

L. 466-477 – The optimal conditions should be reported in Results section. Here condition tested for the method optimization.

L. 507 – Was this method original? Reference?

L. 522 - The document  number must be given.

L. 645 – It should be “independent sample t-test”.

L. 670-684 – The information about Nagoya protocol is too long.

L. 722-750 – They are “Results” not “Discussion”.

First part of Conclusions is like as a Summary.

L. 919-923 – The authors’ initials not full names should be used,

References must be in the MDPI journal style (journal title abbreviations, paper titles not with bold, after  volume must be “,” instead of “:”..

DOI must be added to References.

Reviewer 3 Report

 I. the paper requires restructuring: eg materials and methods before results.

the resolution of the figures is very low which makes the analysis impossible

the article is too complex which makes it difficult to read. 4. the language needs to be reviewed 5. the abstract is not clear and don’t summurized properly the paper 6-the introduction is too long and poorly structured. 7- the paper contains several non-useful information and missing information (ex: the sources of the materials used for the networking study is missed) 8- Missed and not well cited references 9- Missed statistical analysis of table 16 II. the chemical analysis is very poorly presented and analyzed III. Explanation of doses used IV. The presented spectra of the extracts are unclear and not true.    

Round 2

Reviewer 2 Report

The authors corrected this paper properly taken under considerations all my comments. Therefore, I can accept it now.

Author Response

Thank you for your approval of the manuscript.

Reviewer 3 Report

1. English has to be improve.

2. The figures and tables titles text size are not homogeneous ( you should put the titles in the same police and size, please refer to the template) 
homogenize the size of the text inside the tables.

3. Correct figure 6 (the KEGG enrichment map has missing information down the map)

4. The results observation of 2.2.2. Quantitative analysis of amygdalin in the kernel of P. mira is missing

5. The figure 8 chromatograms are not clear please improve the resolution

6. The figure 9 structures are not clear put better quality images (better from ChemDraw) and avoid putting the names (using the numeric system is better) and cite them on the text

7. Please correct the title "material and methods"

8. I Have asked you to put the material and methods part first, you have too many results it's better to introduce the experiment to make reading your article more comfortable.

9. I have asked you to put the results and discussion together so you can make observations, interpretations, and discussions with literature for each result.
